# Identification of protein-protected mRNA fragments and structured excised intron RNAs in human plasma by TGIRT-seq peak calling

Jun Yao[†], Douglas C Wu[†], Ryan M Nottingham, Alan M Lambowitz*

Institute for Cellular and Molecular Biology and Departments of Molecular Biosciences and Oncology, University of Texas, Austin, United States

**Abstract** Human plasma contains > 40,000 different coding and non-coding RNAs that are potential biomarkers for human diseases. Here, we used thermostable group II intron reverse transcriptase sequencing (TGIRT-seq) combined with peak calling to simultaneously profile all RNA biotypes in apheresis-prepared human plasma pooled from healthy individuals. Extending previous TGIRT-seq analysis, we found that human plasma contains largely fragmented mRNAs from > 19,000 protein-coding genes, abundant full-length, mature tRNAs and other structured small non-coding RNAs, and less abundant tRNA fragments and mature and pre-miRNAs. Many of the mRNA fragments identified by peak calling correspond to annotated protein-binding sites and/or have stable predicted secondary structures that could afford protection from plasma nucleases. Peak calling also identified novel repeat RNAs, miRNA-sized RNAs, and putatively structured intron RNAs of potential biological, evolutionary, and biomarker significance, including a family of full-length excised intron RNAs, subsets of which correspond to mirtron pre-miRNAs or agotrons.

*For correspondence:
lambowitz@austin.utexas.edu

[†]These authors contributed equally to this work

## Introduction

Extracellular RNAs in human plasma have been avidly pursued as potential biomarkers for cancer and other human diseases (*Schwarzenbach et al., 2011*; *Akat et al., 2014*; *Rapisuwon et al., 2016*; *Tsang et al., 2017*; *Cheung et al., 2019*; *Pardini et al., 2019*; *Sole et al., 2019*). In healthy individuals, plasma RNAs arise largely by cell death or secretion from cells in blood, bone marrow, lymph nodes, and liver, while in cancer and other diseases, plasma RNAs may arise by necrosis or secretion from tumors or other damaged tissues, potentially providing diagnostic information (*Lo et al., 1999*; *Turchinovich et al., 2011*; *Koh et al., 2014*; *Kishikawa et al., 2015*; *Kishikawa et al., 2016*; *Akat et al., 2019*; *Giraldez et al., 2019*). As plasma contains active RNases (*Kamm and Smith, 1972*; *Tsui et al., 2002*), the extracellular RNAs that persist there are thought to be protected from degradation by bound proteins, RNA structure, or encapsulation in extracellular vesicles (EVs). Although high-throughput RNA-sequencing (RNA-seq) has identified virtually all known RNA biotypes in human plasma, studies aimed at identifying disease biomarkers have focused mostly on plasma mRNAs or miRNAs. mRNAs in plasma and blood have been profiled by RNA-seq methods that enrich for or selectively reverse transcribe poly(A)-containing RNAs (*Koh et al., 2014*; *Englert et al., 2019*) or by sequencing mRNA fragments with (*Freedman et al., 2016*; *Yeri et al., 2017*; *Akat et al., 2019*; *Giraldez et al., 2019*) or without (*Qin et al., 2016*; *Tsang et al., 2017*) size selection, and it remains unclear which methods might be optimal for biomarker identification. miRNAs in plasma are analyzed by methods that enrich for small RNAs and neglect pre-miRNAs or longer transcripts (*Mitchell et al., 2008*; *Turchinovich et al., 2011*; *Blondal et al., 2013*; *Williams et al., 2013*; *Yeri et al., 2017*). Additionally, almost all RNA-seq studies of plasma RNAs

have used retroviral reverse transcriptases (RTs) to convert RNAs into cDNAs for sequencing on high-throughput DNA sequencing platforms. Retroviral RTs have inherently low fidelity and processivity and even those that are highly engineered have difficulty reverse transcribing through stable RNA secondary structures or post-transcriptional modifications, resulting in under-representation of 5'-RNA sequences and aborted reads that can be mistaken for RNA fragments (*Nottingham et al., 2016*). Thus, we still have an incomplete understanding of the biology of plasma RNAs and how to optimize their identification as biomarkers for human diseases.

As a substitute for retroviral RTs, we have been developing RNA-seq methods using thermostable group II intron reverse transcriptases (TGIRTs) (*Mohr et al., 2013*; *Nottingham et al., 2016*; *Qin et al., 2016*). In addition to high fidelity, processivity, and strand-displacement activity, group II intron RTs have a proficient template-switching activity that enables efficient, seamless attachment of RNA-seq adapter sequences to target RNAs without RNA tailing or ligation (*Mohr et al., 2013*; *Lentzsch et al., 2019*). Further, unlike retroviral RTs, which tend to dissociate from RNAs at post-transcriptional modifications that affect base pairing, TGIRT enzymes pause at such modifications but eventually read through by characteristic patterns of mis-incorporation that can be used to identify the modification (*Katibah et al., 2014*; *Nottingham et al., 2016*; *Qin et al., 2016*; *Li et al., 2017*; *Safra et al., 2017*; *Shurtleff et al., 2017*; *Zubradt et al., 2017*). This combination of activities enables TGIRT enzymes to give relatively uniform 5'- and 3'-sequence coverage of mRNAs when initiating conventionally from an annealed oligo(dT) primer (*Mohr et al., 2013*) and to give full-length, end-to-end reads of tRNAs and other structured small ncRNAs, when initiating by template switching to the 3' end of the RNA (*Mohr et al., 2013*; *Katibah et al., 2014*; *Nottingham et al., 2016*; *Qin et al., 2016*). In size selected RNA preparations, TGIRT-seq profiles human miRNAs with bias equal to or less than alternative methods (*Mohr et al., 2013*; *Xu et al., 2019*).

The comprehensive TGIRT-seq method used in this work to analyze human plasma RNAs employs TGIRT-template-switching for 3'-RNA-seq adapter addition followed by a single-stranded DNA ligation for 5' RNA-seq adapter addition (*Figure 1A*). In a validation study using rRNA-depleted, chemically fragmented human reference RNAs with External RNA Control Consortium spike-ins, TGIRT-seq gave better quantitation of mRNAs and spike ins, more uniform 5' to 3' coverage of mRNA sequences, detected more splice junctions, particularly near the 5' ends of mRNAs, and had higher strand specificity when compared to benchmark TruSeq-v3 datasets (*Nottingham et al., 2016*). This study also showed that TGIRT-seq enables the simultaneous sequencing of chemically fragmented mRNAs together with tRNAs and other structured sncRNAs, which were poorly represented in the TruSeq datasets, even after chemical fragmentation (*Nottingham et al., 2016*). A subsequent study of chemically fragmented human cellular RNAs with customized spike-ins confirmed these findings and showed that TGIRT-seq enables simultaneous quantitative profiling of mRNA fragments and sncRNAs of > 60 nt, but under-represents smaller RNAs due in part to differential loss during library clean-up to remove adapter dimers (*Boivin et al., 2018*). Recent TGIRT-seq profiling of structured RNAs in unfragmented cellular RNA preparations from human cells revealed previously unannotated sncRNAs, including novel snoRNAs, tRNA-like RNAs, and tRNA fragments (tRFs) (*Boivin et al., 2020*).

The ability of TGIRT-seq to simultaneously profile mRNAs and non-coding RNAs from small amounts of starting material without size selection is advantageous for the analysis of extracellular RNAs, which are present in low concentrations in human plasma or EVs secreted by cultured human cells. In a previous study, TGIRT-seq showed that human plasma from a healthy male individual contained predominantly full-length tRNAs and other structured small ncRNAs, which could not be seen in RNA-seq studies using retroviral RTs, together with RNA fragments derived from large numbers of protein-coding genes and lncRNAs, with higher proportions of intron and antisense RNAs than in cellular RNA datasets (*Qin et al., 2016*). Similarly, TGIRT-seq showed that highly purified EVs and exosomes secreted by cultured human cells also contained predominantly full-length tRNAs, Y RNAs, and Vault RNAs, together with low concentrations of mRNAs, including 5' terminal oligopyrimidine (5' TOP) mRNAs (*Shurtleff et al., 2017*). The same TGIRT-seq workflow using template switching for facile 3'-RNA-seq adapter enabled single-stranded (ss) DNA-seq profiling of human plasma DNA, including analyses of nucleosome positioning and DNA methylation sites that inform tissue-of-origin (*Wu and Lambowitz, 2017*), as shown previously for other ssDNA-seq methods (*Chandrananda et al., 2015*; *Snyder et al., 2016*).

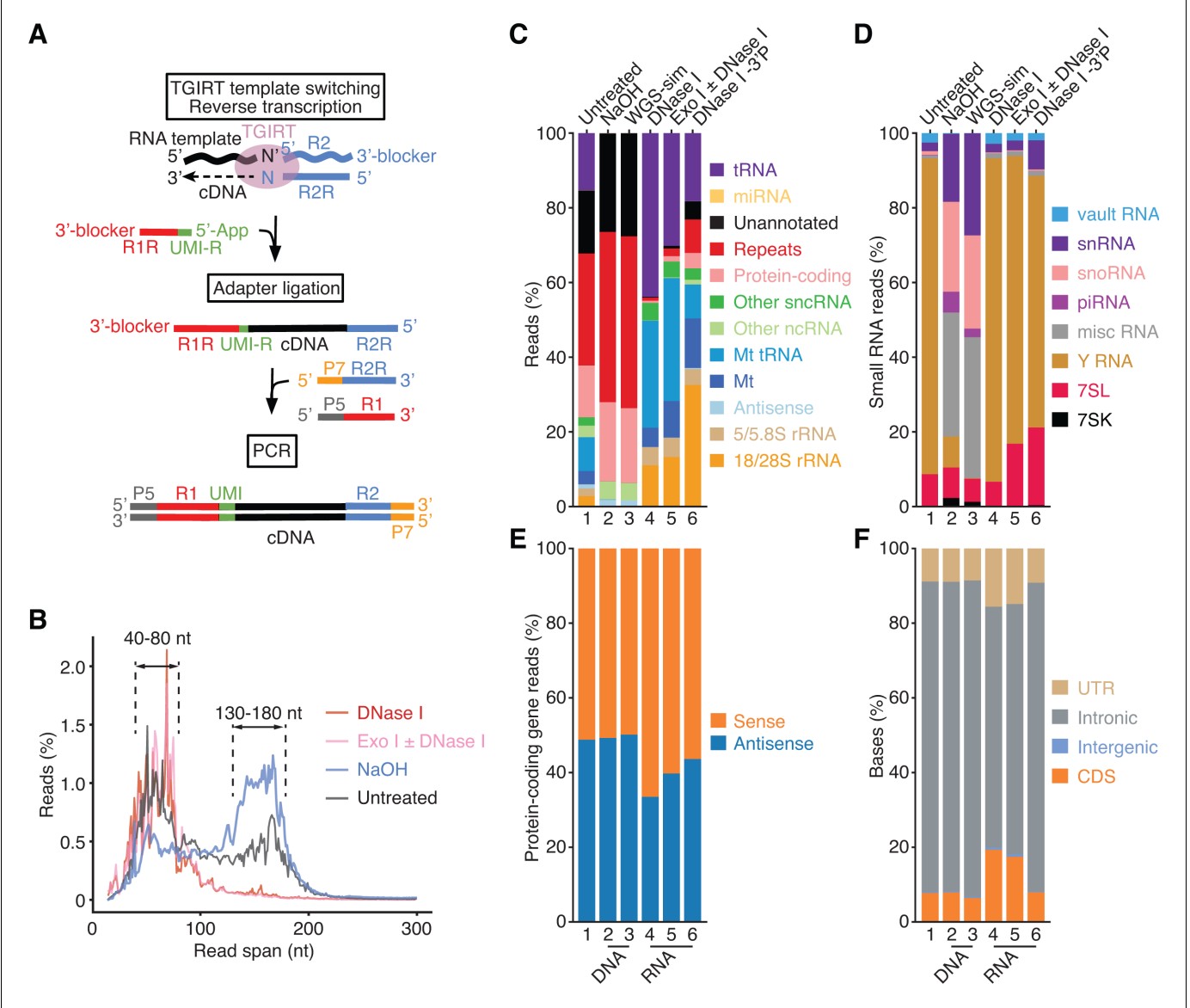

**Figure 1.** TGIRT-seq of nucleic acids in human plasma. (A) TGIRT-seq workflow. The TGIRT enzyme initiates reverse transcription by template switching from a synthetic RNA template/DNA primer substrate consisting of a 35-nt RNA oligonucleotide containing an Illumina read 2 (R2) sequence annealed to a 36-nt DNA primer. The latter leaves a single nucleotide 3' overhang (N, an equimolar mix of A, C, G, and T) that directs template switching by base pairing to the 3' nucleotide (N') of an RNA or DNA template, which is then reverse transcribed resulting in a DNA copy of the nucleic acid with the reverse complement of an Illumina read two sequence (R2R) seamlessly linked to its 5' end (**Qin et al., 2016**; **Wu and Lambowitz, 2017**). After clean-up, a second adapter (R1R DNA), which contains the reverse complement of an Illumina read one sequence and introduces a 6-nt UMI (denoted UMI-R for the reverse orientation), is ligated to the 3' end of the cDNA using thermostable 5' App DNA/RNA ligase, followed by PCR with primers that add capture sites and indices for Illumina sequencing. (B) Read-span distributions in combined datasets for plasma nucleic acids before (n = 3) and after treatment with NaOH to digest RNA (n = 4) or with DNase I (n = 12) or Exonuclease I (Exo I) ± DNase I (n = 3) to digest DNA. The plot shows the percentage of read spans as a function of length calculated from the starting positions of each deduplicated 2 × 75 nt paired-end read. (C) and (D) Stacked bar graphs showing the percentage of deduplicated read pairs in the indicated combined datasets that mapped to different categories of annotated genomic features or unannotated regions of the hg19 human genome reference sequence compared to a simulated dataset of 2 × 75 nt paired-end reads (WGS-sim) generated computationally from the hg19 sequence. Genomic features follow Ensembl annotations. Mt indicates mitochondrial transcripts, and Repeats indicates Repbase-annotated repeats. Other ncRNA includes pseudogenes and Ensembl-annotated long non-coding RNAs (lncRNAs). Other sncRNA indicates the small non-coding RNAs classified by Ensembl and broken out separately in panel (D). (E) Stacked bar graphs showing the percentages of reads that mapped to the sense or antisense strand of protein-coding genes. Strand specificity was calculated after filtering out read pairs that aligned to embedded sncRNAs, such as snoRNAs, but included read pairs that aligned to repeats within protein-coding genes. (F) Stacked bar graphs showing the percentage of bases that mapped to different regions of the sense strand of protein-coding genes. UTR, 5'- or 3'-untranslated region; CDS, coding sequences; intergenic, regions upstream or downstream of transcription start and stop sites annotated

*Figure 1 continued on next page*

*Figure 1 continued*

in RefSeq. In panels (C)-(F), features are color-coded as indicated in the Figure, and numbers at the bottom indicate the same samples with different categorizations.

The online version of this article includes the following figure supplement(s) for figure 1:

**Figure supplement 1.** Bioanalyzer traces of plasma nucleic acids before and after various treatments.
**Figure supplement 2.** Correlation matrix for individual datasets of plasma RNA after different DNA-removal treatments: DNase I (n = 12) and Exo I ± DNase I (n = 3).
**Figure supplement 3.** IGV screenshots showing examples of full-length mature tRNAs and tRNA fragments detected in plasma by TGIRT-seq.
**Figure supplement 4.** IGV screenshots showing examples of full-length mitochondrial (Mt) tRNAs and tRNA fragments.
**Figure supplement 5.** IGV screenshots showing examples of sncRNAs detected in combined datasets for DNase I-treated plasma RNA samples (n = 12).
**Figure supplement 6.** Human plasma contains different miRNA isoforms.

Since these earlier studies, TGIRT-seq methods have undergone improvements including: (i) the use of modified RNA-seq adapters that substantially decrease adapter-dimer formation; (ii) computational and biochemical methods for remediating residual end biases, enabling profiling of miRNAs with accuracy equivalent to the least biased current methods; and (iii) modified reaction conditions that increase the efficiency of RNA-seq adapter addition by TGIRT template switching, enabling the more efficient capture of target RNAs (*Lentzsch et al., 2019*; *Xu et al., 2019*).

Here, we used an updated TGIRT-seq method incorporating these improvements together with RNA-seq adapters that add a unique molecular identifier (UMI) to deconvolute duplicate reads to comprehensively profile RNAs present in commercial human plasma from healthy individuals prepared by apheresis, a method used to obtain large volumes of plasma for clinical purposes, including convalescent plasma from recovered COVID-19 patients (*Duan et al., 2020*). Additionally, we compared the fragmented mRNAs detected in plasma by TGIRT-seq with the polyadenylated mRNAs detected in the same plasma by ultra-low input SMART-Seq v4 and introduced the use of a peak-calling algorithm for analyzing TGIRT-seq datasets. We thus identified numerous mRNA fragments corresponding to annotated binding sites for ~100 different RNA-binding proteins. We also identified a wide variety of discrete structured RNAs and RNA fragments, including abundant repeat and transposable element RNAs, unannotated miRNA-sized RNAs, and intron RNAs. The latter included a family of putatively structured full-length, excised intron RNAs, some corresponding to mirtron pre-miRNAs and/or agotrons, and putatively structured intron RNA fragments, including a family corresponding to conserved structured segments of retrotransposed mRNAs that inserted within long introns.

## Results

### TGIRT-seq of human plasma nucleic acids

The TGIRT-seq datasets in this study were obtained from commercial human plasma pooled from multiple healthy individuals and prepared by apheresis with EDTA as the anticoagulant (IPLA-N-K2E; Innovative Research). For each dataset, nucleic acids were extracted from 4 mL of plasma by using a QIAamp ccfDNA/RNA kit (Qiagen) and treated with NaOH to obtain RNA-free plasma DNA (n = 4) or with DNase I (n = 12), exonuclease I (Exo I; n = 1) or Exo I + DNase I (n = 2) to obtain plasma RNA with minimal residual DNA (referred to collectively as DNase-treated plasma RNA; n = 15). Each preparation gave ~10 ng of nucleic acid, which yielded ~8 ng DNA or ~2 ng RNA, as judged by Bioanalyzer analysis (*Figure 1—figure supplement 1*). Although the QIAamp ccfDNA/RNA kit gave a lower yield of plasma RNA than other kits, it provided a uniform method for obtaining both RNA and DNA from the same plasma preparations.

To prepare TGIRT-seq libraries, we used the workflow outlined in *Figure 1A*, which is based on the previously described TGIRT total RNA-seq method (*Nottingham et al., 2016*; *Qin et al., 2016*) with the following improvements. First, the initial TGIRT-template switching reaction was done at a lower salt concentration (200 instead of 450 mM), which increases the efficiency of 3'-RNA-seq adapter addition (*Wu and Lambowitz, 2017*; *Lentzsch et al., 2019*). Second, we used a modified R2R adapter that substantially decreases adapter-dimer formation (*Xu et al., 2019*), thereby

improving the representation of very small RNAs, such as miRNAs and short tRNA fragments (tRFs). Finally, we used a modified R1R adapter with a 6-nt UMI to deconvolute duplicate reads (*Wu and Lambowitz, 2017*). The libraries were sequenced on an Illumina NextSeq 500 instrument to obtain 10 to 27 million 75-nt paired-end reads, which were mapped to the hg19 human genome reference sequence. Mapping rates for the DNase-treated plasma RNA datasets (n = 15) ranged from 82–96% (*Supplementary file 1*), with pairwise Pearson correlation coefficients r = 0.67–1.00 (*Figure 1—figure supplement 2*). Although, we were concerned that the lower salt concentration used for template switching might increase the frequency of multiple sequential template switches, the percentages of soft-clipped and fusion/discordant read pairs, which include reads from multiple template switches, were relatively low (1.5–2.5% and 0.5–4.1%, respectively; *Supplementary file 1*).

The read-span distributions for the untreated and NaOH-treated plasma DNA samples showed two broad peaks, one at 130–180 nt, corresponding to plasma DNA protected in chromatosomes (nucleosomes plus linker histone) or trimmed mononucleosomes, and the other at 50–100 nt corresponding to nicked DNA strands that are protected by transcription factors or other bound proteins (*Figure 1B*), in agreement with previous studies (*Chandrananda et al., 2015*; *Snyder et al., 2016*; *Wu and Lambowitz, 2017*). Bioanalyzer traces using a High Sensitivity DNA kit showed that all of the dsDNA in the untreated plasma nucleic acid sample was sensitive to DNase I (*Figure 1—figure supplement 1A*). The read-span distributions for plasma RNA samples obtained after treatment with DNase I (n = 12) or Exo I ± DNase I (n = 3) showed a broad peak of 40–80 nt (*Figure 1B*), which corresponded to a similarly sized peak that was detected and confirmed to be NaOH-sensitive in Bioanalyzer traces obtained by using an RNA 6000 Pico Kit (*Figure 1—figure supplement 1B*).

## Overview of RNA biotypes detected in human plasma

*Figure 1C–F* show stacked bar graphs comparing the proportion of read pairs mapping to different genomic features in combined datasets obtained from plasma nucleic acids before and after different treatments. For NaOH-treated plasma DNA samples (n = 4), the overall distribution of read pairs was similar to that simulated for random sampling of human genomic DNA (WGS-sim; simulated by ART). Most of these plasma DNA read pairs mapped to genomic repeats (46%; Repbase), unannotated sequences (26%), or protein-coding genes (21%; *Figure 1C*), with the read pairs mapping to protein-coding genes equally distributed between the sense and antisense strands (*Figure 1E*) and most of the bases in these read pairs located within introns (*Figure 1F*).

The plasma RNA samples after treatment with DNase I (n = 12), Exo I ± DNase I (n = 3), or DNase I + T4 polynucleotide kinase to remove 3' phosphates (n = 3) gave read pairs mapping to 26,000–43,000 different genes (*Figure 1C*, *Figure 1—figure supplement 2* and *Supplementary file 1*). For the DNase I-treated plasma RNA, high proportions of the read pairs mapped to cellular or mitochondrial (Mt) tRNAs (44% and 29%, respectively) followed by cellular and Mt rRNAs (16% and 5%, respectively), other sncRNAs (4.6%), repeats (Repbase; 0.8%), protein-coding gene transcripts (0.6%), other ncRNAs (0.1%), and miRNAs (0.006%; *Figure 1C*). Most of the rRNA reads corresponded to 18S and 28S rRNA (69%), with the remainder to 5S and 5.8S rRNAs (31%; *Figure 1C*). Treatment of plasma nucleic acids with T4 polynucleotide kinase to remove 3' phosphates increased the proportions of total reads mapping to 18S and 28S rRNAs (to 33%), repeats (to 9%), and protein-coding genes (to 4.1%; *Figure 1C*; lane 6). In agreement with previous findings (*Qin et al., 2016*), the proportions of protein-coding gene reads mapping to the antisense strand (17–36%; *Figure 1E*) and bases within introns (63–73%; *Figure 1F*) were higher than those in TGIRT-seq datasets of cellular RNAs (10% and 24%, respectively [*Nottingham et al., 2016*]), suggesting that intron and antisense RNAs may be selectively exported or stable in plasma.

The most abundant sncRNAs in the DNase I-treated plasma RNA samples were tRNAs (36,766 counts per million (CPM) for Val-CAC, the most abundant isodecoder species) followed by Y RNAs (15,247 CPM for RNY4, the most abundant species), and 7SL RNA (158–1,504 CPM for the three 7SL RNA genes; *Figure 1D* and *Supplementary file 1*). As found previously (*Qin et al., 2016*), TGIRT-seq showed that most of the cellular and Mt tRNAs detected in plasma were full-length mature tRNAs, which cannot be fully reverse transcribed by retroviral RTs (*Figure 1—figure supplements 3* and *4*, respectively). The plasma preparations also contained lower concentrations of specific cellular 5'-tRNA halves (0.02–68 CPM), 3'-tRNA halves (0.02–9,036 CPM), and shorter tRNA fragments (5'-tRFs, 0.02–39 CPM; 3'-tRFs, 3–640 CPM; and internal 2-tRFs, 0.02–7 CPM), with the proportion of reads corresponding to some 5'-tRNA halves and 5'-tRFs increasing after 3'-phosphate removal

(*e.g.*, 5' half Gly-CCC, 2.9-fold increase; 5'-tRF Cys-GCA, 2-fold increase; see *Supplementary file 1*). Notably, TGIRT-seq identified 5' ends of 3'-tRFs that extend beyond the $m^1A$ modification site in the TΨC loop, which is a strong stop for retroviral RTs and can be mistaken for the 5' end of 3'-tRFs (*Figure 1—figure supplement 3*; *Kumar et al., 2014*; *Reinsborough et al., 2019*). Other structured sncRNAs in plasma, including Vault RNAs, snoRNAs, snRNAs, 7SK RNA, 7SL RNA, 5S rRNA and 5.8S rRNA, were also present largely as full-length mature transcripts (*Figure 1—figure supplement 5*), in agreement with previous findings (*Qin et al., 2016*).

The improved TGIRT-seq method detected 263 miRNAs annotated as high confidence miRNAs in miRbase v20 in the DNase I-treated plasma RNAs datasets (n = 12; *Supplementary file 1*). As found previously using TGIRT-seq (*Qin et al., 2016*), these were a mixture of pre- and mature miRNA, passenger strands, and longer transcripts, whose proportions varied for different miRNA species (*Figure 1—figure supplement 6*). The most abundant miRNA detected by TGIRT-seq, miR-223, was present at only 2.89 CPM (*Figure 1—figure supplement 6*), considerably lower than the most abundant similarly sized 3'-tRFs (Glu-TTC, 640 CPM; Glu-CTC, 510 CPM; Lys-CTT, 121 CPM; *Supplementary file 1*). Only a very low proportion of reads (< 0.001%) mapped to piRNA loci (*Figure 1D*; lanes 4–6), and none corresponded to a mature piRNA, even though under the lower salt conditions used for library construction TGIRT-III can template switch to and reverse transcribe RNAs with 2'-O-methylated 3' ends characteristic of piRNAs (*Lentzsch et al., 2019*).

## Identification of bacterial and viral RNA and DNA reads in plasma prepared by apheresis

To identify bacterial and viral RNAs in the commercial plasma preparations, we used the taxonomic classification program Kraken2 to analyze non-human reads in combined DNase-treated plasma RNA datasets (n = 15; 3.3 million deduplicated reads including reads mapping to *E. coli*, which were filtered from the datasets and reintroduced for this analysis; see Materials and methods). Kraken2 assigns reads to bacterial, viral, and archaeal genomes based on matches to unique *k*-mer sequences in a database of non-redundant bacterial, viral, and archaeal sequences compiled from RefSeq. We thus identified 2.83 million RNA reads assigned to bacteria, with the most abundant phyla being Proteobacteria, Actinobacteria, and Firmicutes and the most abundant genera being *Pseudomonas*, *Salmonella*, *Acinetobacter*, *Halomonas*, and *Bacillus*, consistent with other studies (*Potgieter et al., 2015*; *Pan et al., 2017*). A much smaller number of non-human RNA reads (~52,000) were assigned to viruses, the most abundant being *Parvoviridae* and Mason-Pfizer monkey virus, a retrovirus (*Figure 2A*). The combined NaOH-treated plasma DNA datasets (n = 4; 46 million deduplicated reads) contained only 12,000 non-human reads, much lower than 1% reported in another study using plasma prepared by a different method (*Kowarsky et al., 2017*). Most (93.3%) of the non-human DNA reads were assigned by Kraken2 to bacteria, mainly *Pseudomonas* spp., with only 64 reads assigned to viruses, mostly human β herpesvirus 6B (*Figure 2B* and *Supplementary file 1*). The commercial plasma preparations had been screened by the supplier for hepatitis B virus, hepatitis C virus, HIV-1, HIV-2, and syphilis, and we found no reads mapping to any of these pathogens. A separate search for Epstein-Barr virus (EBV) found 28 deduplicated RNA reads and no DNA reads. Although EBV can reside in latent form in B lymphocytes, it was detected by qPCR in only 30% of plasma samples from individuals without active EBV disease (*Gulley and Tang, 2008*). More generally, the low levels of viral nucleic acids found here could also reflect that apheresis-prepared plasma has very low levels of leukocyte and platelet contamination, the presence of which correlates with increased risk of viral and prion disease transmission in plasma used for clinical purposes (*Valbonesi et al., 2001*).

## TGIRT-seq versus ultra low input SMART-Seq v4

Our previous TGIRT-seq analysis of plasma from a healthy male individual, which was prepared from freshly drawn blood by sedimentation into a Ficoll-cushion to minimize cell breakage, revealed that the mRNAs in plasma are largely fragmented, consistent with other studies of plasma RNAs using different RNA-seq methods with or without size selection (*Freedman et al., 2016*; *Tsang et al., 2017*; *Yeri et al., 2017*; *Akat et al., 2019*; *Giraldez et al., 2019*). In the plasma prepared by apheresis from multiple healthy individuals, the proportion of the TGIRT-seq reads mapping to protein-coding genes was relatively low (0.6–4.1%; *Figure 1C*, lanes 4–6), but nevertheless sufficient to

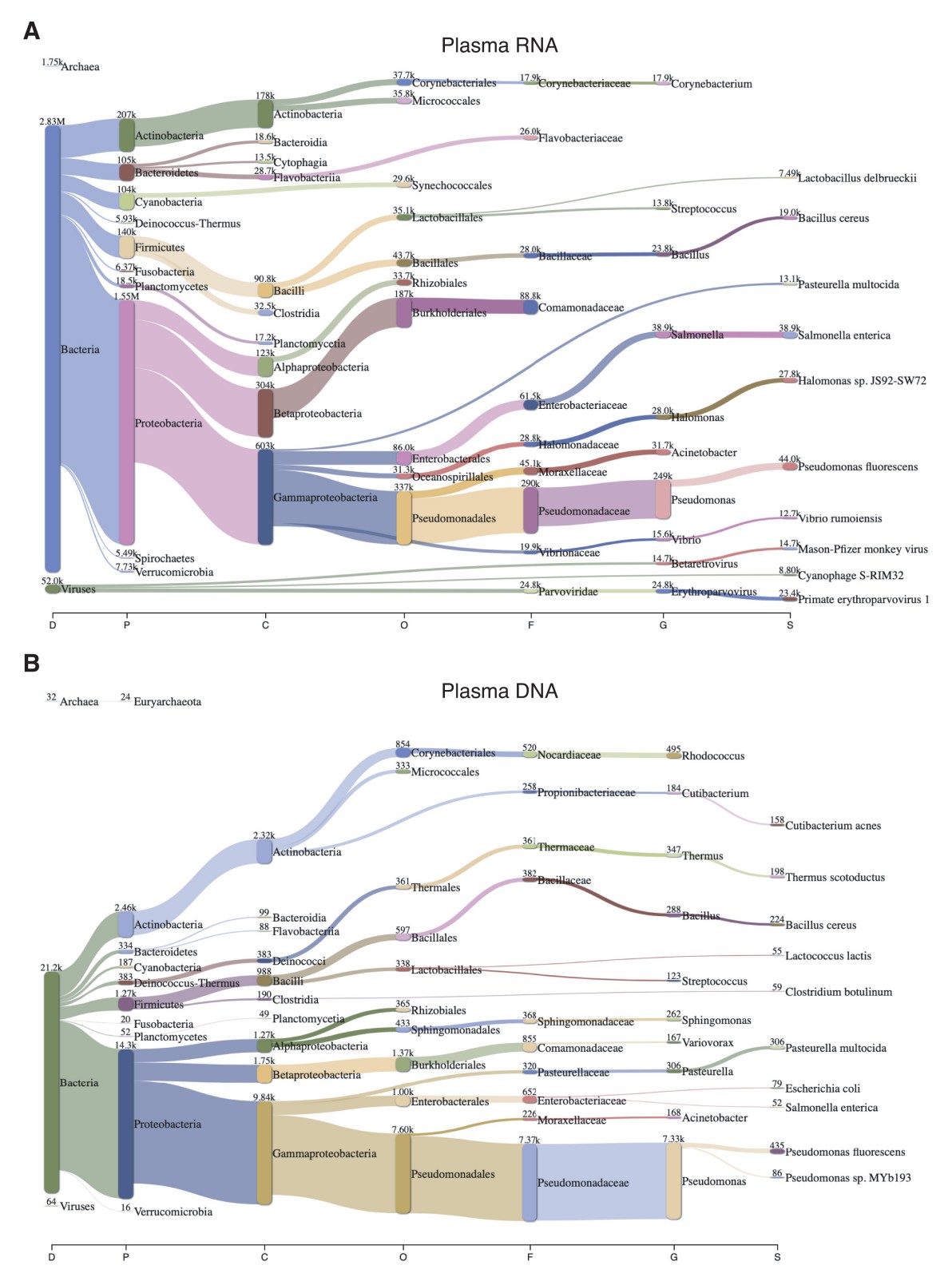

**Figure 2.** Sankey diagram of reads assigned to bacteria, archaea, and viruses from combined datasets of (**A**) DNase-treated plasma RNA (n = 15) and (**B**) NaOH-treated plasma DNA (n = 4). The diagram shows the flow of mapped reads from the most general (left) to most specific (right) taxonomic classification (abbreviations: D, Domain; P, Phylum; C, Class; O, Order; F, Family; G, Genus; S, Species). The width of the flow is proportional to the number of reads, which is indicated at the top of each node.

identify mRNAs originating from 15,000 to 19,000 different protein-coding genes, with the highest number of different mRNAs detected after 3'-phosphate removal with T4 polynucleotide kinase (*Figure 3A and B*). The identities and relative abundances of different protein-coding gene transcripts in the apheresis-prepared plasma were broadly similar to those in the previous TGIRT analysis of plasma prepared by Ficoll-cushion sedimentation of blood from a healthy male individual (*Qin et al., 2016*) ($r$ = 0.62–0.80; *Figure 3C*) and between high quality plasma samples similarly prepared from five healthy females in a collaborative study with Dr. Naoto Ueno, M.D. Anderson ($r$ = 0.53–0.67; manuscript in preparation).

To investigate the relationship between the mRNAs detected in plasma by TGIRT-seq and the polyadenylated mRNAs detected by other methods (*e.g.*, *Koh et al., 2014*; *Englert et al., 2019*), we compared the protein-coding gene transcripts identified in the TGIRT-seq datasets for DNase I-treated plasma RNAs (n = 12) with those identified in identically prepared RNA from the same plasma preparations by ultra low input SMART-Seq v4 (Takara Bio; n = 4; *Figure 3D–E* and *Figure 3—figure supplements 1* and *2*). The latter is a highly sensitive method for profiling polyadenylated mRNAs in which an engineered retroviral RT (SMARTScribe) initiates cDNA synthesis from an anchored oligo(dT) primer with an appended PCR primer-binding site and then template switches from the 5' end of that mRNA to an acceptor oligonucleotide containing a second primer-binding site, enabling PCR amplification of the resulting cDNAs. In our experiments, the resulting dsDNAs were fragmented by Covaris sonication and used to prepare DNA-seq libraries by using an NEBNext Ultra II DNA Library Prep (New England Biolabs). The SMART-Seq libraries (n = 4) were then sequenced on an Illumina NextSeq 500 to obtain a total of 2.6 million 2 × 75 nt reads that mapped to > 16,000 protein-coding genes with a transcript per million value (TPM) > 0.1. The larger number of mRNA reads compared to TGIRT-seq (0.28 million) largely reflects that SMART-Seq selectively profiles polyadenylated mRNAs, while TGIRT-seq profiles mRNAs together with other more abundant RNA biotypes. In addition, ultra low input SMART-Seq is not strand-specific, resulting in redundant sense and antisense strand reads (*Figure 3—figure supplement 1*). Bioanalyzer traces showed that the amplified dsDNAs obtained by SMART-Seq prior to Covaris sonication were substantially shorter than those from the cellular RNA control provided with the kit (peaks at 100–200 and 400–700 bp for plasma RNA compared to 700–7,000 bp for the cellular control RNA; *Figure 3D*), indicating that the poly(A)-containing mRNAs detected in plasma by SMART-Seq are enriched in 3' fragments.

A scatter plot comparing the relative abundance of transcripts originating from different genes showed that most of the polyadenylated mRNAs detected in DNase I-treated plasma RNA by ultra low input SMART-Seq were also detected by TGIRT-seq at similar TPM values when normalized for protein-coding gene reads ($r$ = 0.61), but with some, mostly lower abundance mRNAs undetected either by TGIRT-seq or SMART-Seq, and with SMART-Seq unable to detect non-polyadenylated histone mRNAs, which are relatively abundant in plasma (*Figure 3E* and *Figure 3—figure supplement 1*). Similar correspondences were found for TGIRT-seq of plasma RNAs after 3'-phosphate removal (n = 3; $r$ = 0.58) or after additional chemical fragmentation followed by 3'-phosphate removal (n = 4; $r$ = 0.68; *Figure 3—figure supplement 2A and B*). Heat maps comparing protein-coding gene transcripts detected by TGIRT-seq and SMART-Seq to primary tissue and platelet transcriptome data (*Schubert et al., 2014*; *Uhlén et al., 2015*; *Campbell et al., 2018*) indicated that the detected mRNA originated largely from hematopoietic tissues, including bone marrow, lymph nodes, and spleen, with contributions from erythrocytes, white blood cells, and platelets (*Figure 3—figure supplement 2C and D*), in agreement with other studies (*Koh et al., 2014*; *Giraldez et al., 2019*). The highly represented mRNAs detected by both TGIRT-seq and SMART-Seq included hemoglobins and other blood cell mRNAs (*e.g.*, pro-platelet basic protein (PPBP) and S100 calcium-binding proteins S100A8 and S100A9), which are involved in regulating immune responses (*Ryckman et al., 2003*; *Figure 3E* and heat maps *Figure 3—figure supplement 2D*). Other prominent mRNA families detected by both methods included 5' TOP mRNAs, which encode ribosomal proteins and other components of the translational apparatus and are translationally repressed by recruitment to RNP granules under stress conditions (*Damgaard and Lykke-Andersen, 2011*) and aminoacyl-tRNA synthetase (aaRS) mRNAs, which have potential intercellular signaling associated with different protein isoforms generated by alternative splicing (*Wang et al., 2013*; *Figure 3* and *Figure 3—figure supplement 2*). Both 5' TOP and aaRS mRNAs have also been found in EVs secreted by cultured human

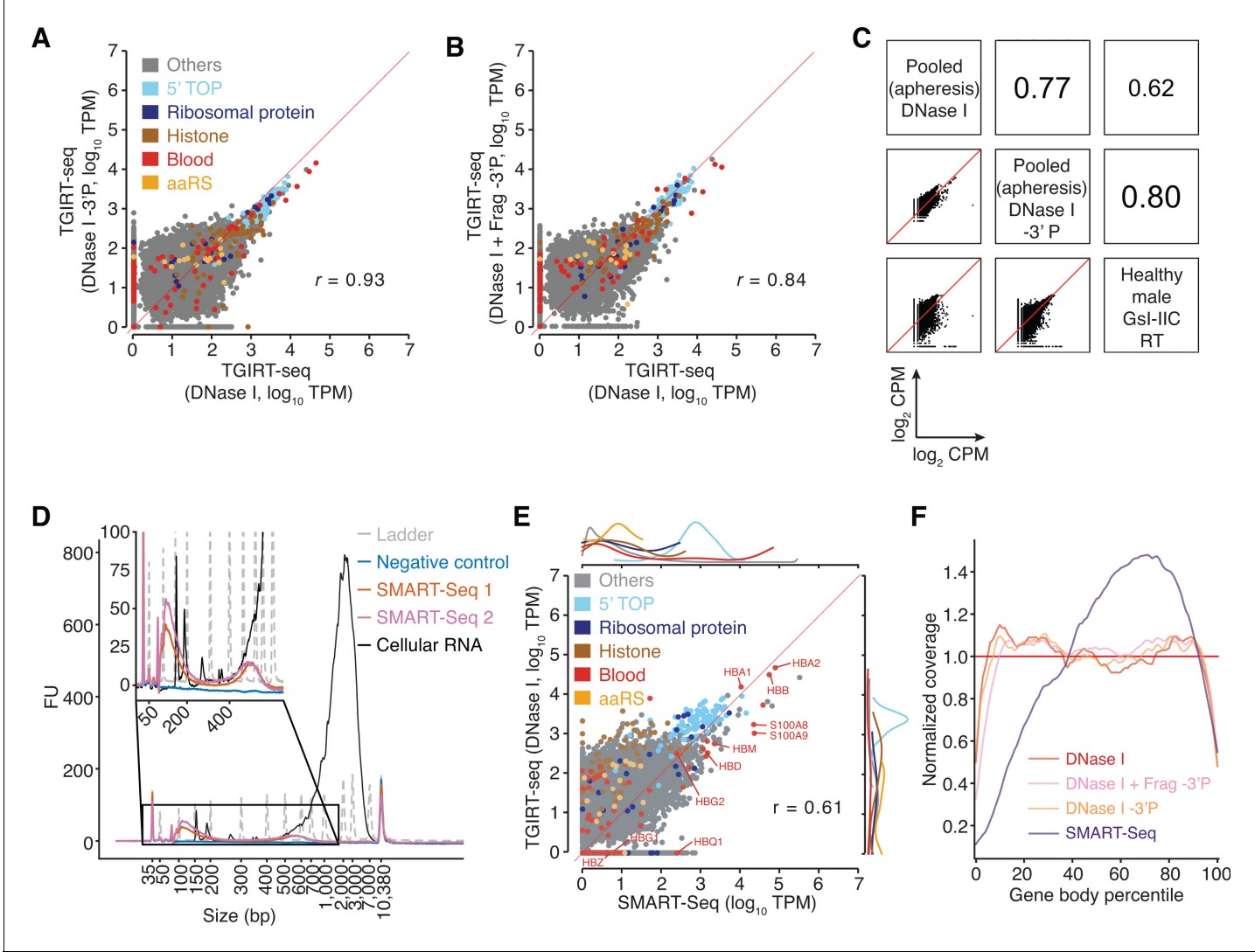

**Figure 3.** TGIRT-seq analysis of protein-coding gene transcripts in human plasma and comparison with SMART-Seq. (A, B) Scatter plots comparing protein-coding gene transcripts detected by TGIRT-seq in combined datasets for DNase I-treated plasma RNA without (n = 12) or with (n = 3) 3'-phosphate removal or chemical fragmentation followed by 3'-phosphate removal (n = 5). Reads aligned to protein-coding genes were extracted and quantified by Kallisto to obtain transcript-per-million (TPM) values, and average TPM values for each gene were plotted for each pairwise comparison. Each point represents one gene and is color-coded by type of encoded protein as indicated in the Figure. 5' TOP includes those ribosomal proteins whose mRNAs have a 5' TOP sequence. The red line indicates identical TPM values for both methods. Pearson correlation coefficients are shown at the lower right. (C) Scatter plots comparing the relative abundances of protein-coding gene transcripts detected in DNase I-treated plasma RNA using the improved TGIRT-seq method in this work without (n = 12) or with (n = 3) 3'-phosphate removal with those detected previously by TGIRT-seq in DNase I-treated plasma RNA from a healthy male individual (n = 3; SRP064378, datasets 12–14) (*Qin et al., 2016*). CPMs for each gene were normalized to the total number of mapped reads in the dataset and log$_2$ transformed for the comparisons. Spearman correlation coefficients are shown in sectors at the upper right. (D) Bioanalyzer traces of PCR-amplified cDNAs generated by SMART-Seq according to manufacturer's protocol. Samples were analyzed using an Agilent 2100 Bioanalyzer with a High Sensitivity DNA kit. Traces are color-coded by sample type as shown in the Figure, with 'Ladder' (dashed gray lines) indicating the DNA ladder supplied with the High Sensitivity DNA kit. The negative control was a SMART-Seq library prepared with water as input, and the cellular RNA control was the positive control RNA supplied with the SMART-Seq kit. SMART-Seq 1 and 2 indicate double-stranded DNAs generated by using the SMART-Seq kit from different samples of DNase I-treated plasma RNA. FU, fluorescence units. (E) Scatter plot comparing protein-coding gene transcripts detected in plasma by TGIRT-seq and SMART-Seq. Reads assigned to protein-coding genes transcripts (0.28 million reads for TGIRT-seq and 2.6 million reads for SMART-Seq) were extracted from combined datasets obtained from DNase I-treated plasma RNA (n = 12 for TGIRT-seq and n = 4 for SMART-Seq) and quantified by Kallisto. The scatter plots compare average TPM values for protein-coding gene transcripts with each point representing one gene, color-coded by the type of encoded protein (see above). The marginal distributions of different color-coded mRNA species in the scatter plot are shown above for SMART-Seq and to the right for TGIRT-seq. The red line indicates identical TPM values for both methods. The Pearson correlation coefficient (*r*) is indicated at the bottom right. (F) Normalized 5'- to 3'-gene body coverage for protein-coding gene transcripts detected in plasma by TGIRT-seq and SMART-Seq. Gene body coverage was computed by Picard tools (Broad Institute) using the genomic

*Figure 3 continued on next page*

*Figure 3 continued*

alignment files generated by Kallisto. The plots show normalized gene coverage versus normalized gene length for all protein-coding transcripts in the indicated datasets color-coded as indicated in the Figure. The red horizontal line at y = 1 indicates perfectly uniform 5' to 3' coverage.

The online version of this article includes the following figure supplement(s) for figure 3:

**Figure supplement 1.** IGV screen shots showing examples of protein-coding gene transcripts detected in DNase I-treated plasma RNA by SMART-Seq and TGIRT-seq.

**Figure supplement 2.** Additional scatter plots and heat maps comparing protein-coding gene transcripts detected in DNase I-treated plasma RNA by TGIRT-seq and SMART-Seq.

cells (*Wang et al., 2013*; *Shurtleff et al., 2017*), a potential vehicle by which these mRNAs might enter plasma.

Plots of normalized coverage over all detected mRNAs showed that SMART-Seq under-represented 5'-RNA sequences and over-represented 3'-RNA sequences, as expected for 3'-mRNA fragments, while TGIRT-seq coverage, with or without 3'-phosphate removal or with chemical fragmentation followed by 3' phosphate removal, was relatively uniform across most of the length of the detected mRNAs (*Figure 3F*). Integrated Genomics Viewer (IGV) alignments for representative protein-coding genes confirmed the relatively uniform coverage of both polyadenylated and non-polyadenylated mRNAs by TGIRT-seq and showed that TGIRT-seq simultaneously detected an intron-encoded snoRNA in a ribosomal protein gene that was invisible to SMART-Seq (*RPL10*; upper right in *Figure 3—figure supplement 1*).

Notably, although 3'-phosphate removal increased both the proportion of reads mapping to protein-coding genes and the number of mRNAs detected by TGIRT-seq (see above), neither the 5'- to 3'-coverage nor the abundance of most protein-coding transcripts detected in plasma by TGIRT-seq was strongly affected (*Figure 3A,B and F*; $r$ = 0.93 and 0.84, respectively). This reflects that many of the additional protein-coding gene transcripts detected after 3'-phosphate removal had relatively low read counts and that the abundance of only a small proportion of protein-coding gene transcripts increased significantly after this treatment (0.41% by differential expression analysis using DESeq2; adjusted p-value < 0.01). Because 3' phosphates inhibit RNA-seq adapter addition by TGIRT template switching (*Mohr et al., 2013*), these findings are consistent with the suggestion from the previous TGIRT-seq analysis (*Qin et al., 2016*) that many mRNA fragments in plasma have 3' OH termini, as expected for RNA fragments generated by cellular ribonucleases that function in mRNA processing or turnover (*Houseley and Tollervey, 2009*; *Schoenberg and Maquat, 2012*).

## Use of peak-calling for analysis of plasma RNAs

As DNA fragments that persist in plasma were found to be packaged in nucleosomes or associated with transcription factors or other DNA binding proteins that afford protection from plasma nucleases (*Chandrananda et al., 2015*; *Snyder et al., 2016*; *Wu and Lambowitz, 2017*), we wondered whether many of the mRNA fragments in plasma might be similarly protected by bound proteins and whether such protein-protected RNA fragments could be detected by peak calling, as done for ChIP-seq (*Zhang et al., 2008*). To test this idea, we combined the TGIRT-seq datasets for all DNase-treated plasma samples (n = 15) and removed reads from human blacklist regions, which are known to produce artifactual peaks (*Amemiya et al., 2019*), as well as reads corresponding to rRNAs, Mt RNAs, and annotated sncRNAs, which were analyzed above. We were left with ~1.6 million dedupli-cated read pairs that mapped to the hg19 human genome reference sequence to use as input for the ChIP-seq narrow peak caller, MACS2 (*Zhang et al., 2008*). As a base line control, we used the ~38 million deduplicated read pairs that mapped to human genome from the NaOH-treated plasma DNA samples (n = 4).

Peak calling using a read coverage cutoff of $\geq$ 5, a false discovery rate cutoff of 0.05 (q-value assigned by MACS2), and a requirement that the peak was detected in at least 5 of the 15 samples to avoid batch effects yielded 1,036 peaks that were enriched in the DNase-treated plasma RNA over the base line control. After further filtering to remove read alignments with MAPQ < 30 (a cut-off that eliminates reads mapping equally well at more than one locus) or $\geq$ 5 mismatches from the mapped locus, we were left with 950 high confidence peaks ranging in size from 59 to 1,207 nt with $\geq$ 5 high quality read alignments at the peak maximum (*Supplementary file 1*). The percentage

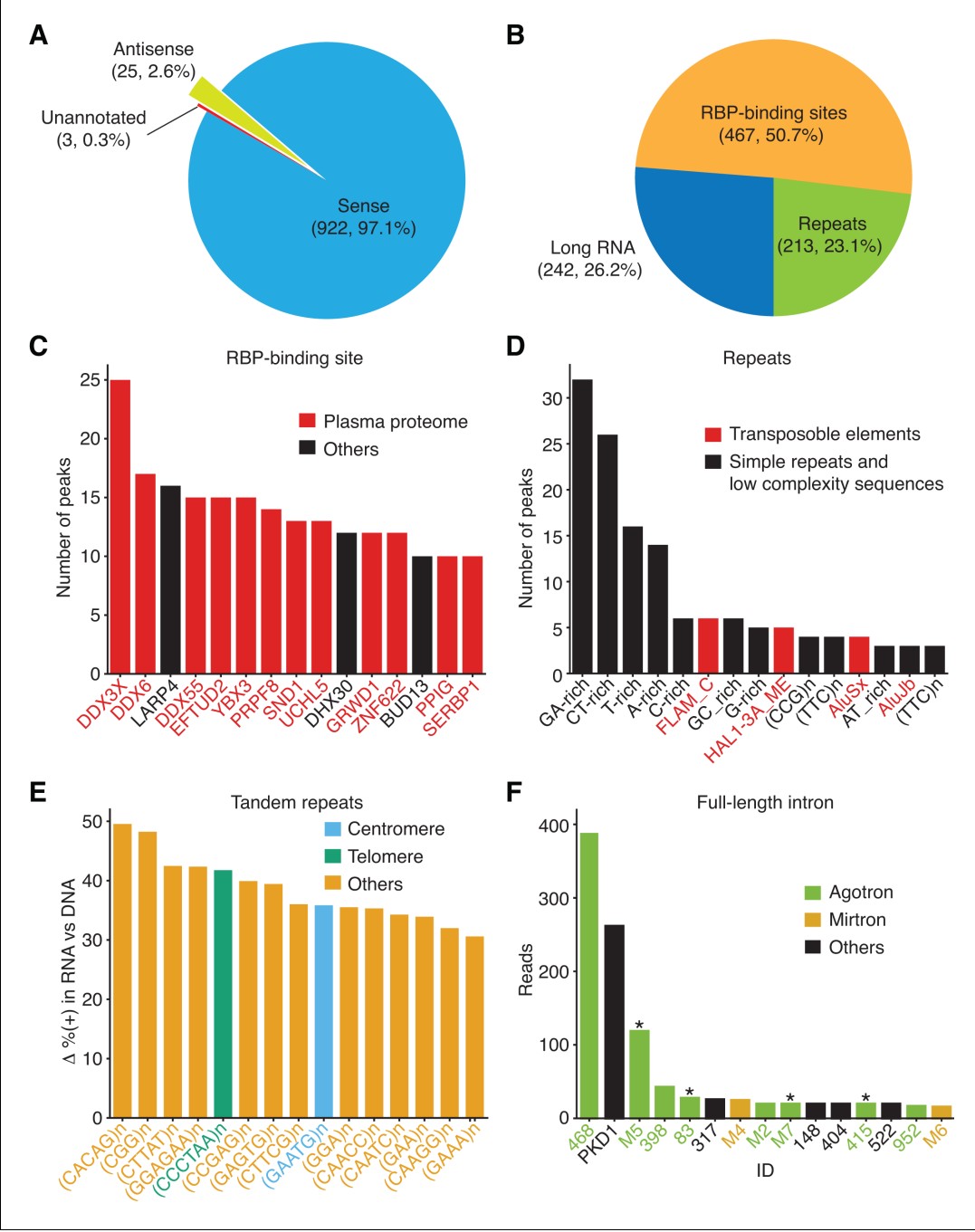

**Figure 4.** Peak-calling analysis of plasma RNA. The MACS2 algorithm was used to call peaks in combined TGIRT-seq datasets for DNase-treated plasma RNA (n = 15) after removal of the reads for rRNAs, Mt RNAs, and annotated sncRNAs (tRNAs, Y RNAs, 7SK and 7SL RNAs, vault RNAs, snoRNAs, snRNAs, high confidence miRNAs, and miscellaneous RNAs). The combined datasets for NaOH-treated plasma DNA (n = 4) were used as the base-line control. Only peaks with q-value < 0.05 assigned by MACS2 and supported by alignments from at least five separate libraries at the peak summit were used for the analysis. Peaks were intersected sequentially with (i) binding sites for 109 RNA-binding protein (RBPs) from ENCODE K-562 and HepG2 cell eCLIP datasets; (ii) RefSeq and GENCODE gene annotations; and (iii) repeat annotations from Repbase, and a best feature annotation for each peak was selected by the highest overlap score, as described in Materials and methods. (**A**) and (**B**) Pie charts showing the proportion of peaks with annotations on the sense strand, antisense strand, or no annotations (**A**) and the proportion of peaks mapping to different categories of genomic features with sense-strand annotations (**B**). The number of peaks in each sector and their percentage of the total are indicated in parentheses. (**C–F**) Bar

*Figure 4 continued on next page*

*Figure 4 continued*

graphs showing the (C) top 15 RBP-binding sites ranked by peak count, with RBPs found to be part of the plasma proteome (*Nanjappa et al., 2014*) shown in red; (D) top 15 Repbase annotated repeats ranked by peak count, with simple repeats and low complexity sequences in black and transposable element RNAs in red; (E) top 15 short tandem repeat RNAs with highest difference (Δ) in the percentage of (+) orientation reads between the DNase-treated plasma RNA and NaOH-treated plasma DNA datasets color coded as indicated in the Figure; (F) top 15 peaks corresponding to full-length excised intron RNAs ranked by the number of deduplicated reads. The peak IDs are indicated on the x-axis and are color-coded by annotated intron RNA type: agotron (green); mirtron (gold); annotated as both agotron and mirtron (green with asterisk). Others, not annotated in the previous categories (black). The read count for *PKD1* is the combined count from the peak in *PKD1* gene and the peaks for the same intron in six *PKD1* pseudogenes (*Figure 9D* below and peak IDs #335–340).

The online version of this article includes the following figure supplement(s) for figure 4:

**Figure supplement 1.** Characteristics of different categories of peaks identified by MACS2 peak calling analysis of combined datasets of DNase-treated plasma RNA (n = 15).
**Figure supplement 2.** Overlap, quality control, and statistical significance of MACS2-called peaks corresponding to annotated RBP-binding sites.
**Figure supplement 3.** IGV screenshots showing peaks mapping to unannotated genomic loci.

and number of peaks mapping to different annotated features are shown by pie charts in *Figure 4A and B*. Among the detected peaks, 922 had one or more annotations for a gene (Ensemble), RBP-binding site (ENCODE K-562 and HepG2 cells eCLIP datasets [*Van Nostrand et al., 2016*]), or repeat sequence (Repbase) on the sense strand; 25 had one or more such annotations on the anti-sense strand; and three had no overlap with any of these annotations.

To understand the origins of these peaks, we first extracted those with sense-strand annotations and plotted the proportions corresponding to different annotated genomic features (*Figure 4B*). About half of the peaks (467 peaks; 50.7%) contained or overlapped an ENCODE eCLIP annotated RBP-binding site in a long RNA (*i.e.*, a mRNA, lncRNA, or pseudogene RNA). A smaller proportion (242 peaks; 26.2%) mapped to long RNAs but did not contain an ENCODE eCLIP-annotated RBP-binding site, and the remainder (213 peaks; 23.1%) were genomic repeats, including simple sequence repeats and transposable element RNAs. The distributions of lengths and read depths for different categories of peaks are shown in (*Figure 4—figure supplement 1*).

## RNA fragments containing RBP-binding sites are prevalent in plasma

The 467 peaks with an ENCODE eCLIP annotated RBP-binding site contained or overlapped the binding sites for 90 of the 109 RBPs searched initially in the eCLIP datasets, including 75 that had been shown previously to be part of the plasma proteome (*Nanjappa et al., 2014*; *Supplementary file 1*). These 467 peaks comprised 0.8% of the long RNA reads (1.4% of the long RNA reads with MAPQ $\geq$ 30; *Figure 4—figure supplement 2A*); ranged in size from 64 to 823 nt (*Figure 4—figure supplement 1A*); and had 11 to 282 nt overlaps with the identified RBP-binding sites (*Figure 4—figure supplement 2B*). Simulations using 1,000 sets of 950 peaks with the same size distribution as the 950 called peaks randomly generated from the sequences of the genes encoding long RNAs detected in plasma indicated that the enrichment of RBP-binding site sequences in the 467 identified peaks was statistically significant (p < 0.001; *Figure 4—figure supplement 2C*).

To determine which RBP-binding sites might be prevalent in plasma, we counted the number of peaks containing or overlapping a called binding site for the same RBP (*Supplementary file 1*). Because some peaks contained annotated binding sites for more than one RBP, for purposes of this count, we assigned the peak to the best matched RBP (or RBPs in case of ties) scored by the number of overlapping bases. The top 15 RBP-binding sites based on the number of assigned peaks are shown in *Figure 4C*, and IGV coverage plots and alignments for representative peaks are shown in *Figure 5*. The six RBPs with the most prevalent binding sites in plasma RNAs based on the number of assigned peaks included: three DEAD-box RNA helicases, DDX3X (*Figure 5A*), DDX6, and DDX55; YBX3, a homologue of the low specificity RBP YBX1, which is not included in the ENCODE eCLIP datasets but of interest because it binds 5′ TOP mRNAs and functions in sorting sncRNAs into exosomes (*Figure 5B*; *Evdokimova et al., 2001*; *Shurtleff et al., 2017*; (iii) LARP4, a protein that

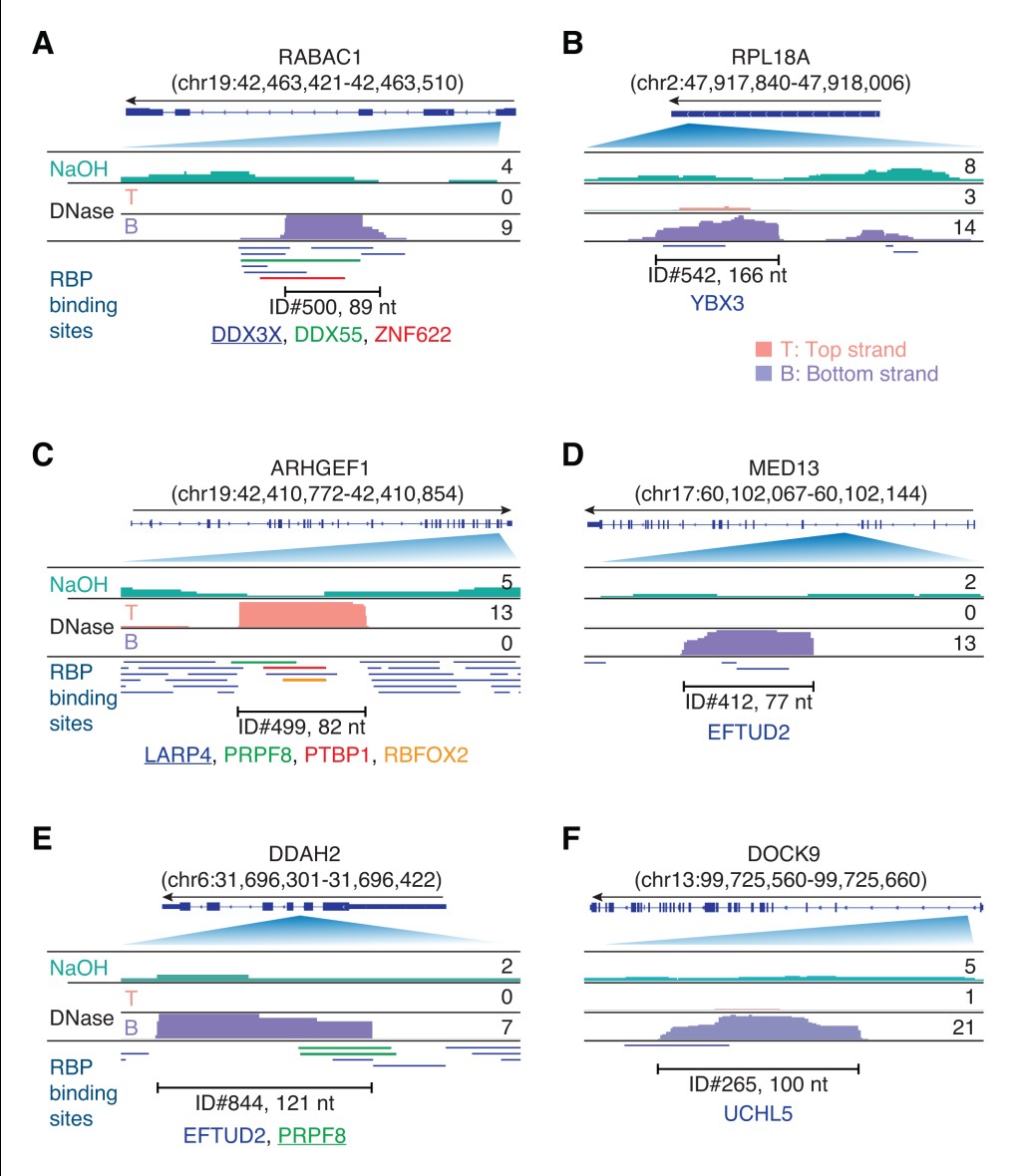

**Figure 5.** Integrative Genomics Viewer (IGV) screenshots showing examples of plasma RNA peaks corresponding to annotated RBP-binding sites detected in plasma by TGIRT-seq peak calling. (A–F) Gene names are indicated at the top with the hg19 coordinates of the called peak in parentheses and an arrow below indicating the 5' to 3' orientation of the encoded RNA. The top track shows the gene map (exons, thick bars; intron, thin lines), with the relevant part of the gene map expanded below. The tracks below the gene map show coverage plots for the peak region in combined datasets for NaOH-treated plasma DNA (n = 4; combined top and bottom strands, turquoise) or DNase-treated plasma RNA (n = 15; Top strand (T), pink and Bottom strand (B), purple) based on the number of deduplicated reads indicated to the right. The peak called by MACS2 is delineated by the bracketed line, with the peak ID and length indicated below. Annotated protein-binding sites are shown as color-coded lines below the coverage plots. For cases in which the peak overlaps multiple RBP-binding sites, the one whose annotated binding site had the largest number of bases overlapping the peak is underlined.

regulates translation *Figure 5C*); (iv) and EFTUD2, an RNA splicing factor (*Figure 5D*). Farther down the list are PRPF8, a spliceosomal protein that provides a structural scaffold for the assembly of snRNAs at splice sites (*Figure 5E*); SND1, a transcriptional co-activator; and UCHL5, an RBP that modulates mRNA expression (*Figure 5F*). Also prevalent in plasma were RNA fragments containing binding sites for DHX30, a DExH-box RNA helicase; GRWD1, a histone-binding protein that may also play a role in ribosome assembly; ZNF622, a zinc-finger protein; BUD13, a spliceosome

component; PPIG, a protein involved in protein-folding; and SERBP1, a mRNA-binding-protein. Notably, seven of these top 15 proteins (DDX3X, DDX6, DHX30, LARP4, SERBP1, SND1, and YBX3) are components of stress granules (*Markmiller et al., 2018*; *Nunes et al., 2019*).

While most of the potential RBP-binding sites identified by peak calling against the human genome reference sequence corresponded to fragments of mRNA exons (409 binding sites), pseudogenes transcripts (10 binding sites), or annotated antisense RNAs (two binding sites), 46 mapped to introns, and surprisingly, 29 corresponded to short, putatively structured, full-length excised intron RNAs (*Supplementary file 1*). These full-length excised intron RNAs corresponded to annotated binding sites for 13 different RNA-binding proteins, including five spliceosomal proteins (7 **PRPF8**, 5 **EFTUD2**, 3 XRN2, 3 XPO5, 2 **SF3A3**, 2 **SF3B4**, and one each for DDX24, HNRNPM, LARP4, LIN28B, **RBM22**, RPS5, and TROVE2; spliceosomal proteins in bold). The remaining 17 intron RNA peaks mapped within longer introns and corresponded to annotated binding sites for a largely different set of 13 RBPs (3 YBX3, 2 ILF3 and TRA2A, and one each **EFTUD2**, GPKOW, METAP2, SAFB2, SF3B1, SLTM, SUB1, TIA1, UCHL5, and ZNF622), with a single occurrence of a spliceosomal protein EFTUD2 binding site among the intron fragments being the only RBP common to both sets of peaks (*Supplementary file 1*). The characteristics of the intron RNAs in these peaks are described below along with those of additional intron peaks that were not annotated as an RBP-binding site.

Peak calling against the human genome reference sequence might miss RBP-binding sites that are close to or overlap exon junctions, as such reads were treated by MACS2 as long reads that span the intervening intron. To address this possibility, we mapped the reads in the combined DNase-treated datasets (n = 15) to a human transcriptome reference sequence (Ensemble human cDNA references) and obtained ~0.26 million deduplicated read pairs that were used as input for MACS2 without a control dataset (*Zhang et al., 2008*). Using the same peak-calling and filtering parameters as above, we identified 806 high confidence peaks (denoted by peak IDs beginning with T), the large majority of which (638 peaks, 79%) were identical or overlapped the 467 RBP-binding site peaks identified by mapping to the human genome reference sequence. Of the 168 newly identified peaks, 135 (80 in mRNAs, 15 in pseudogenes, 11 in antisense RNAs, eight in lincRNAs, and 21 in other lncRNAs) overlapped annotated binding sites for 57 RBPs, only five of which were not identified by peak calling against the genome sequence (AKAP8L, FXR1, RBFOX2, SMNDC1, and TARDBP; *Supplementary file 1*). Forty-seven of these newly identified RBP-binding sites peaks overlapped two or more exons (45 in mRNAs, one in a pseudogene, and one in a lncRNA), with binding sites for AKAP8L (A-Kinase Anchoring Protein 8 Like, which functions in mRNA export) and TARDBP (TAR DNA Binding Protein, which regulates alternative splicing) found exclusively in such peaks (*Supplementary file 1*).

As expected for different populations of mRNAs in plasma and cells, scatter plots showed that the relative abundances of the different RBP-binding sites in plasma RNAs differed markedly from those in the ENCODE eCLIP cellular RNA datasets (without or with the inclusion of peaks mapping to the human transcriptome and both for the original annotation for 109 different RBPs and annotation with irreproducible discovery rate (IDR) analysis from the ENCODE website for 150 different RBPs; *Figure 6* and *Figure 6—figure supplement 1*). The scatter plots identified a number of RBPs whose binding sites were more highly represented in plasma than cellular RNAs or vice versa. Notably, stress granule proteins (bold letters) comprised a high proportion of the RBPs whose binding sites were enriched in plasma RNAs, consistent with a previously suggested link between RNP granules, EV packaging, and RNA export (*Shurtleff et al., 2017*; *Temoche-Diaz et al., 2019*) (see Discussion). Although these findings do not prove that the identified RBP was associated with the RNA peak identified in plasma, collectively they suggest that many of the mRNA fragments that persist in plasma are protected from plasma nucleases by bound RBPs. In addition to providing clues to the origin of plasma RNAs and a guide for further analysis, the called peaks identify mRNA regions that are relatively stable in plasma and may be more readily detected than other more labile regions in targeted liquid biopsies.

## Peaks mapping to mRNA exons and pseudogenes

Two hundred and thirteen peaks that mapped to exons or pseudogenes in the human genome reference sequence did not correspond to ENCODE eCLIP-annotated RBP-binding sites or genomic repeats, and an additional 33 such peaks were identified by mapping to the human transcriptome (*Figure 7*, *Figure 7—figure supplements 1–3*, and *Supplementary file 1*). These peaks ranged in

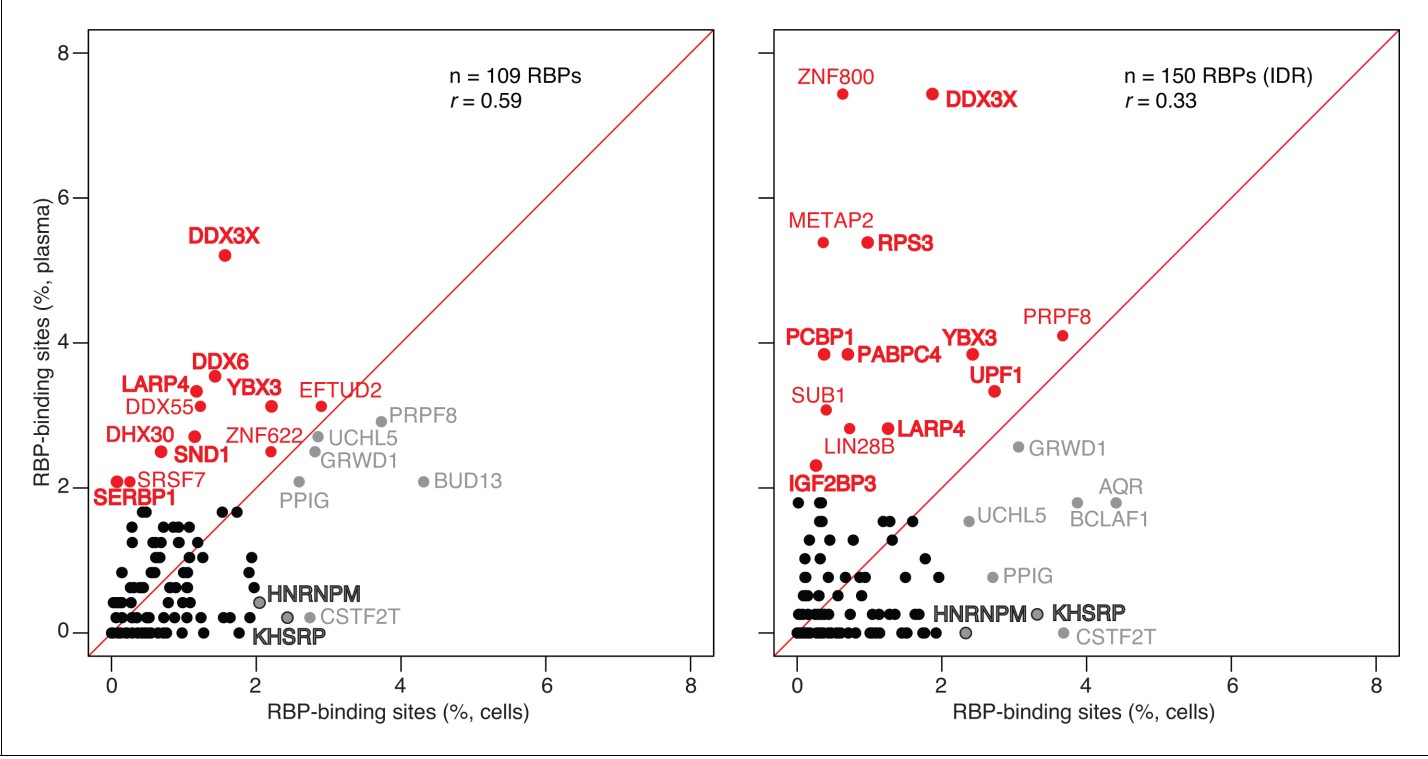

**Figure 6.** Scatter plot comparing the relative abundances of RBP-binding sites in plasma RNA peaks called against the human genome reference sequence with those of RBP-binding sites in ENCODE eCLIP cellular RNA datasets (ENCODE 109 RBP eCLIP dataset on the left and 150 RBP eCLIP dataset with irreproducible discovery rate (IDR) analysis on the right; the RBPs whose binding sites were searched are listed in the *Supplementary file 1*). Abundant RBP-binding sites enriched in plasma or cellular RNAs are indicated in red and gray, respectively. Stress granule proteins are indicated in bold. *r* is the Pearson correlation coefficient.

The online version of this article includes the following figure supplement(s) for figure 6:

**Figure supplement 1.** Scatter plot comparing the relative abundances of RBP-binding sites in plasma RNA peaks called against both the human genome and transcriptome reference sequences with those of RBP-binding sites in ENCODE eCLIP cellular RNA datasets (ENCODE 109 RBP eCLIP dataset on the left and 150 RBP eCLIP dataset with irreproducible discovery rate (IDR) analysis on the right; the RBPs whose binding sites were searched are listed in the *Supplementary file 1*).

size from 52 to 1,207 nt and could correspond to RBP-binding sites that were not included in the searched datasets, unannotated structured sncRNAs, structured regions of mRNAs, or may simply reflect uneven sequence coverage across mRNA sequences.

Twenty five of these 246 peaks were classified by Infernal/Rfam analysis (*Kalvari et al., 2018*) into four types of known RNA structures: 21 histone 3'-UTR stem-loops required for 3'-end processing of histone mRNAs; an iron response element (IRE I) in a ferritin light chain (*FTL*) exon; a selenocysteine insertion sequence 1, which directs the translation of UGA as SelCys; and two potential pre-miRNA stem-loop structures, with the closest match for both being mouse miR-692, which has no annotated human homolog (*Figure 7* and *Supplementary file 1*).

We inspected IGV plots for all of the remaining 221 exon and pseudogene peaks and found that most were comprised of RNA fragments mapping to protein-coding gene exons or pseudogenes (*Figure 7—figure supplements 1–3*). To evaluate whether these peaks contained RNAs that could potentially fold into stable secondary structures, we used RNAfold, a tool that is widely used for this purpose with the understanding that the predicted structures remain to be validated and could differ under physiological conditions or due to interactions with proteins. Of the 221 exon and pseudogene peaks, 213 could be folded by RNAfold into stable RNA secondary structures with a calculated minimum free energy (MFE; $\Delta G$) $\leq -12$ kcal/mol (*Supplementary file 1*). These included 13 peaks that were comprised of or contained reads with discrete 5' and 3' termini (defined as > 50% of reads with the same 5' and 3' termini), a characteristic expected for unannotated sncRNAs (*Figure 7—*

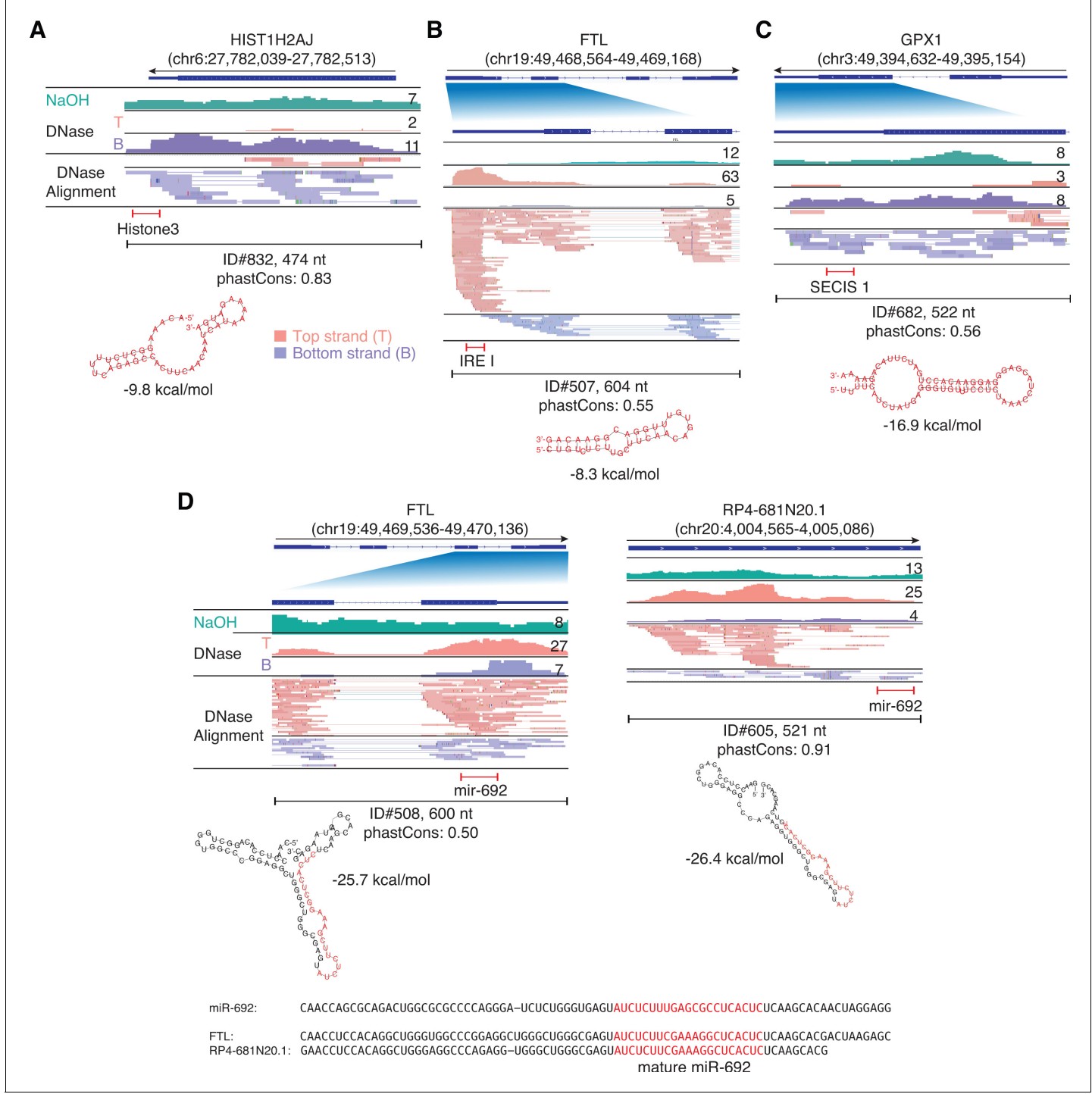

**Figure 7.** IGV screenshots showing examples of peaks mapping to exons or pseudogenes that contain regions classified by Infernal/Rfam (*Kalvari et al., 2018*) as belonging to known classes of structured RNAs. (**A**) A segment of peak near the 3' end of *HIST1H2AJ* mRNA that contains a histone 3'-UTR stem-loop structure. (**B**) A segment of a peak in the 5' UTR of *FTL* (ferritin light chain) mRNA that contains an iron response element I (IRE I). (**C**) A segment of a peak near the 3' end of *GPX1* mRNA that contains a selenocysteine insertion sequence 1 (SECIS 1). (**D**) Segments of peaks near the 3' ends of *FTL* and a pseudogene that are predicted to fold into a pre-miRNA-like stem-loop structure for a miRNA with sequence similarity to mouse miR-692 (sequence alignments shown below). Gene names are indicated at the top with the hg19 coordinates of the called peak in parentheses and an arrow below indicating the 5' to 3' orientation of the encoded RNA. The top track shows the gene map (exons, thick bars; intron, thin lines), with the relevant part of the gene map expanded below if needed to visualize discussed features. The tracks below the gene map show coverage plots and read alignments for the peak region in combined datasets for NaOH-treated plasma DNA (n = 4; combined top and bottom strands, turquoise) or

*Figure 7 continued on next page*

*Figure 7 continued*

DNase-treated plasma RNA (n = 15; Top strand (T), pink and Bottom strand (B), purple) based on the number of deduplicated reads indicated at the right in the coverage tracks. Colors other than pink or purple in the read alignments indicate bases that do not match the reference sequence (red, green, blue, and brown indicate thymidine, adenosine, cytidine, and guanosine, respectively). The peak called by MACS2 is delineated by a bracketed line with the peak ID, length, and phastCons score for 46 vertebrates including humans indicated below (see phylogenetic tree in *Figure 10* below). The bracketed red line indicates the portion of the peak corresponding to the identified feature, whose predicted secondary structure and MFE (ΔG) computed by RNAfold are shown below.

The online version of this article includes the following figure supplement(s) for figure 7:

**Figure supplement 1.** IGV screenshots showing examples of peaks consisting of heterogenous RNA fragments extending across protein-coding gene exons.

**Figure supplement 2.** IGV screenshots showing examples of peaks consisting of heterogenous RNA fragments mapping to pseudogenes.

**Figure supplement 3.** IGV screenshots showing example of peaks mapping to exons or pseudogenes that are comprised of or contain RNA fragments with discrete 5′- and 3′-ends (defined as > 50% of the deduplicated reads having identical 5′ and 3′ ends).

*figure supplement 3A–D*, including four corresponding to unannotated short (14–25 nt) RNAs; *Figure 7—figure supplement 3C and D* and T3803, and T34313, *Supplementary file 1*). Three of the peaks that could not be folded by RNAfold into stable secondary structures were also comprised of or contained reads with discrete 5′ and 3′ ends (*Figure 7—figure supplement 3E–G*), including one containing a relatively abundant unannotated 18-nt RNA (*Figure 7—figure supplement 3E*). Subject to the caveats above regarding conclusions drawn from RNAfold, simulations using peaks randomly generated from long RNA gene sequences indicated that enrichment of RNAs with more stable secondary structures (lower MFEs) in the called RNA peaks was statistically significant (p ≤ 0.019; *Figure 4—figure supplement 2D*).

## Identification of tandem repeats and transposable element RNAs

Although Repbase-annotated sequence repeats constituted only 0.8–2.1% of the reads in the DNase-treated plasma RNA datasets (*Figure 1C*), they comprised almost a quarter of the RNA peaks identified by peak calling (213 peaks; 23.1%), including 154 comprised of or containing short tandem repeats (also referred to as simple repeats or microsatellite sequences) and 59 corresponding to transposable element RNAs (*Figure 4B* and *Supplementary file 1*). TGIRT enzymes are advantageous for the analysis of tandem repeat RNAs because of their ability to reverse transcribe through stable higher-order RNA structures formed by interactions between the repeat units, enabling them to more completely reverse transcribe and better quantitate these RNAs than can retroviral RTs (*Carrell et al., 2018*). The peaks for the short tandem repeat RNAs in human plasma had a surprisingly narrow length distribution (sharp peak ~80 nt) with > 50% having complete or nearly complete read coverage across the annotated repeat region (*Figure 8A and B*). Of the top 15 Repbase-annotated repeat types with the highest peak count, 11 were low complexity sequences, such as polypurine or polypyrimidine tracts or simple repeat RNAs (*e.g.*, (CCG)n and (TTC)n), and four were transposable element RNAs (AluSx, AluJb, FLAM_C and LINE-1 element HAL1-3A_ME) (*Figure 4D* and *Supplementary file 1*). IGV alignments for representative repeat RNAs are shown in *Figure 8—figure supplement 1*.

We noticed that some of the peaks for Repbase simple repeat sequences (*e.g.*, (CATTC)n and (TTAGGG)n) were called by MACS2 at some loci with roughly equal numbers of reads on both strands, possibly reflecting mismapping of DNA or RNA reads. To address this issue, we used an Empirical Bayes method (Materials and methods and *Figure 8C*) to identify the top 15 repeat sequences with the largest differences in the percentage of sense (+) orientation reads between DNase-treated plasma RNA and NaOH-treated plasma DNA and found that all corresponded to tandem repeats of short (3–6 nt) sequences, including centromeric and telomeric repeats (*Figure 4E*, *Figure 8D*, and *Figure 8—figure supplement 1C and D*). We also detected potential HSATII RNAs, which had been identified previously as a repeat RNA present in plasma (*Kishikawa et al., 2016*), but with similar numbers of (+) and (-) orientation reads (2,327 and 2,414, respectively) and not called as peaks by MACS2. Differential expression analysis for transposable element RNAs found that those with highest enrichments in plasma RNA compared to plasma DNA included endogenous retroviral, LINE-1, and Alu RNAs (*Figure 8E*). In many but not all cases, the transposable element peaks corresponded to RNA fragments with peak lengths in the same narrow range found for simple

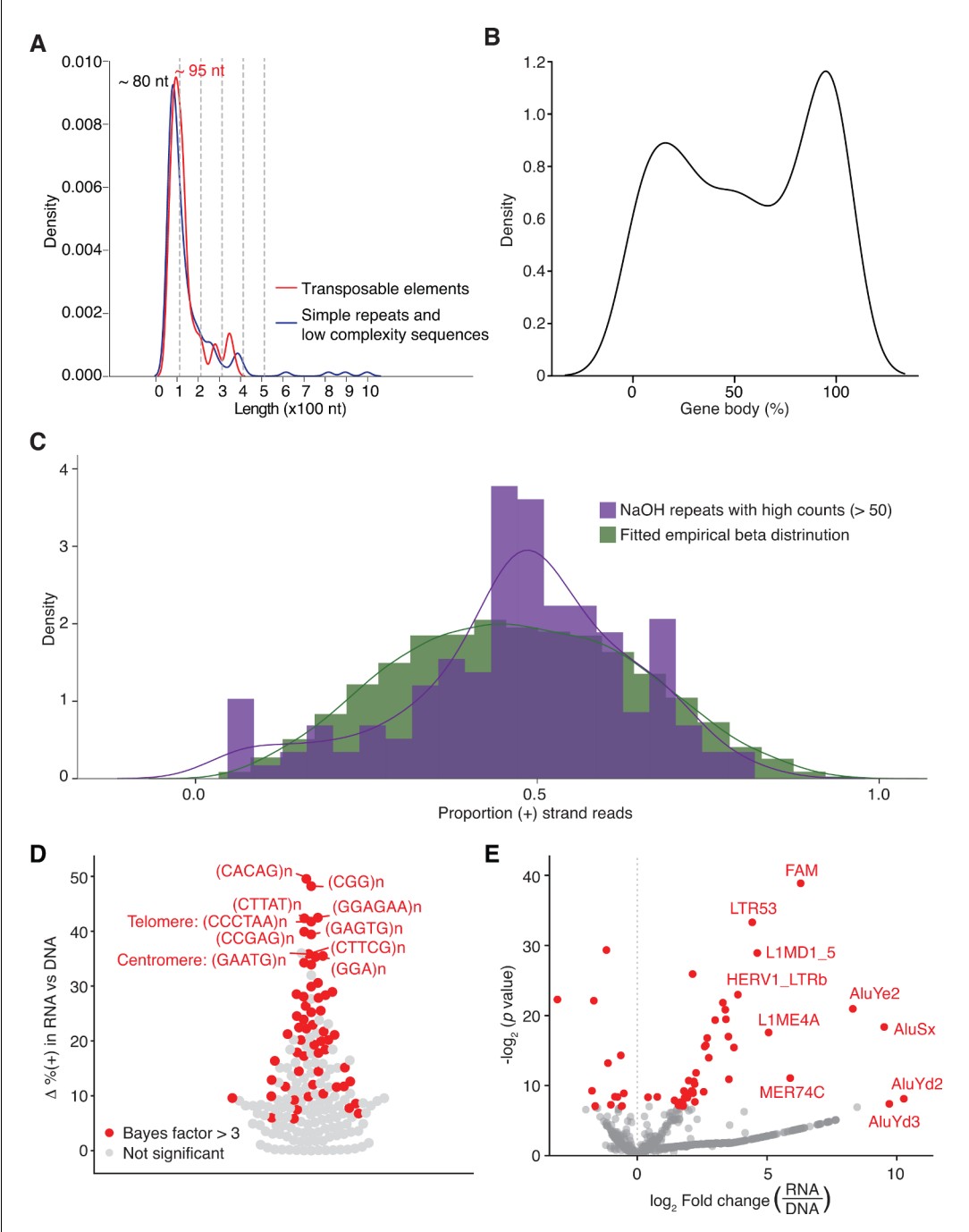

**Figure 8.** Identification and characterization of repeat RNAs in human plasma. (**A**) Density plot showing the range of peak lengths for different types of repeat sequences in the combined datasets for DNase-treated plasma RNA (n = 15). (**B**) Density plot showing the percentage of gene body covered by TGIRT-seq reads for all detected repeat RNAs. The normalized coverage of each repeat sequence in the combined plasma RNA datasets is plotted against the normalized size of the gene (100 bins). (**C**) Fitted beta distribution for the numbers of repeat sequences with different proportions of (+) orientation reads in combined datasets for NaOH-treated plasma DNA datasets (n = 4). The fitted beta distribution was used in an Empirical Bayes method to identify RNA repeats with significant differences in the percentage of (+) orientation reads in the combined plasma RNA datasets compared to the combined plasma DNA datasets, as described in Materials and methods. (**D**) Swarm plot showing the difference (Δ) between the percentage of (+) orientation reads in the combined plasma RNA datasets versus the combined plasma DNA datasets for different short tandem repeat element RNAs. The top ten tandem repeats with the highest (+) orientation enrichments in plasma RNA compared to plasma DNA datasets are named in the plot. (**E**) Volcano plot of fold difference in the normalized read count of transposable elements in DNase-treated plasma RNA (n = 15) compared to NaOH-treated plasma DNA (n = 4). The reads corresponding to repeat sequences in the initial mapping were remapped by SalmonTE to Repbase

*Figure 8 continued on next page*

*Figure 8 continued*

transposable elements, quantified, and compared to assess differences in the normalized read counts in the combined DNase-treated plasma RNA datasets versus the combined NaOH-treated plasma DNA datasets. Each point represents an annotated transposable element, with those in red having a fold-change with an adjusted p-value < 0.05. The top ten transposable elements with the highest fold changes and p-values < 0.05 are named in the plot.

The online version of this article includes the following figure supplement(s) for figure 8:

**Figure supplement 1.** IGV screenshots showing examples of repeat and retrotransposable element RNAs detected in plasma by TGIRT-seq.

repeats (*Figure 8A*, *Figure 8—figure supplement 1G–J*). The detection of short tandem repeat and transposable element RNAs as relatively discrete peaks in plasma could reflect RNA structural features or bound proteins that protect from plasma nucleases.

## Peak calling identifies a family of putatively structured, full-length excised intron RNAs

Twenty-nine peaks identified as containing RBP-binding sites were full-length excised intron RNAs. An additional 14 such peaks were found among those mapping to long RNAs but not containing an annotated RBP-binding site, and another seven (denoted by IDs beginning with M) were not called as peaks but were identified as containing annotated miRNAs sequences. As discussed further below, seven of these 50 intron peaks corresponded to the same intron found in the *PKD1* gene and six of its pseudogenes. Counting these seven peaks as a single intron, we identified 44 different introns for which full-length excised intron RNAs were detected in plasma. The top 15 of these intron RNA peaks ranked by read count are shown in *Figure 4F*, and the complete list is in the *Supplementary file 1*.

All of the full-length excised intron RNAs detected in plasma corresponded to short introns with lengths 73–210 nt (*Supplementary file 1*). All were identified by at least five deduplicated TGIRT-seq reads containing the complete intron sequence beginning with a 5' GU and ending with a 3' AG and could be folded by RNAfold into a stem-loop structures with a minimum free energy (MFE; $\Delta G \leq -18.7$ kcal/mol) (examples shown in *Figure 9A–F*). These intron RNAs are likely linear molecules, as the TGIRT-seq reads begin precisely at the 5' GU and end precisely at the 3' AG with no indication of mismatches or impediments that might be indicative of read through of a branch point in a lariat RNA. In several cases, the reads corresponding to full-length excised intron RNAs had one or two non-templated U or A residues at their 3' end, reminiscent of non-templated 3' A-tails found for yeast linear introns that accumulate in stationary phase cells (*Morgan et al., 2019*).

Thirteen of the full-length excised intron RNAs that we detected in plasma, including the most abundant, *DOCK6* intron 25 (ID#468, 389 deduplicated reads, 7 CPM), corresponded to annotated agotrons, intron RNAs that were identified as binding to Argonaute-2 protein (AGO2) in HITS-CLIP-seq datasets and shown to repress mRNA translation in reporter assays (*Hansen et al., 2016*; *Hansen, 2018*; *Figure 4F*, *Figure 9A* and *Supplementary file 1*). Six other possible agotrons (peak ID#s 2, 404, 416, 522, 844, and 846) were identified by intersecting the full-length excised intron RNAs with AGO1-4 PAR CLIP datasets (*Hafner et al., 2010*). Agotrons had been hypothesized to be full-length linear intron RNAs based on Northern hybridization experiments and CLIP-seq 5'-end sequences (*Hansen et al., 2016*), but to our knowledge, this has not been confirmed previously by RNA-seq, possibly because the retroviral RTs used in other RNA-seq methods are unable to fully reverse transcribe these structured RNAs.

Ten of the excised intron RNAs that we detected in plasma corresponded to annotated mirtrons, miRNAs that are processed by DICER from debranched structured intron RNAs (examples shown in *Figure 9B and C*; *Berezikov et al., 2007*; *Okamura et al., 2007*; *Ruby et al., 2007*). Seven of these ten mirtron pre-miRNAs were also annotated as agotrons (including four of the top 15; green with asterisk in *Figure 4F* and *Supplementary file 1*). All of the mirtron RNA peaks detected in plasma corresponded to the full-length excised intron RNAs (*i.e.*, the putative pre-miRNA), to our knowledge the first time such intron-derived pre-miRNAs have been sequenced, and none corresponded to the annotated mature miRNA.

Significantly, the remaining 21 putatively structured full-length excised intron RNAs, 48% of the total that we detected in plasma, including the second most abundant, *PKD1* intron 29 (ID#333;

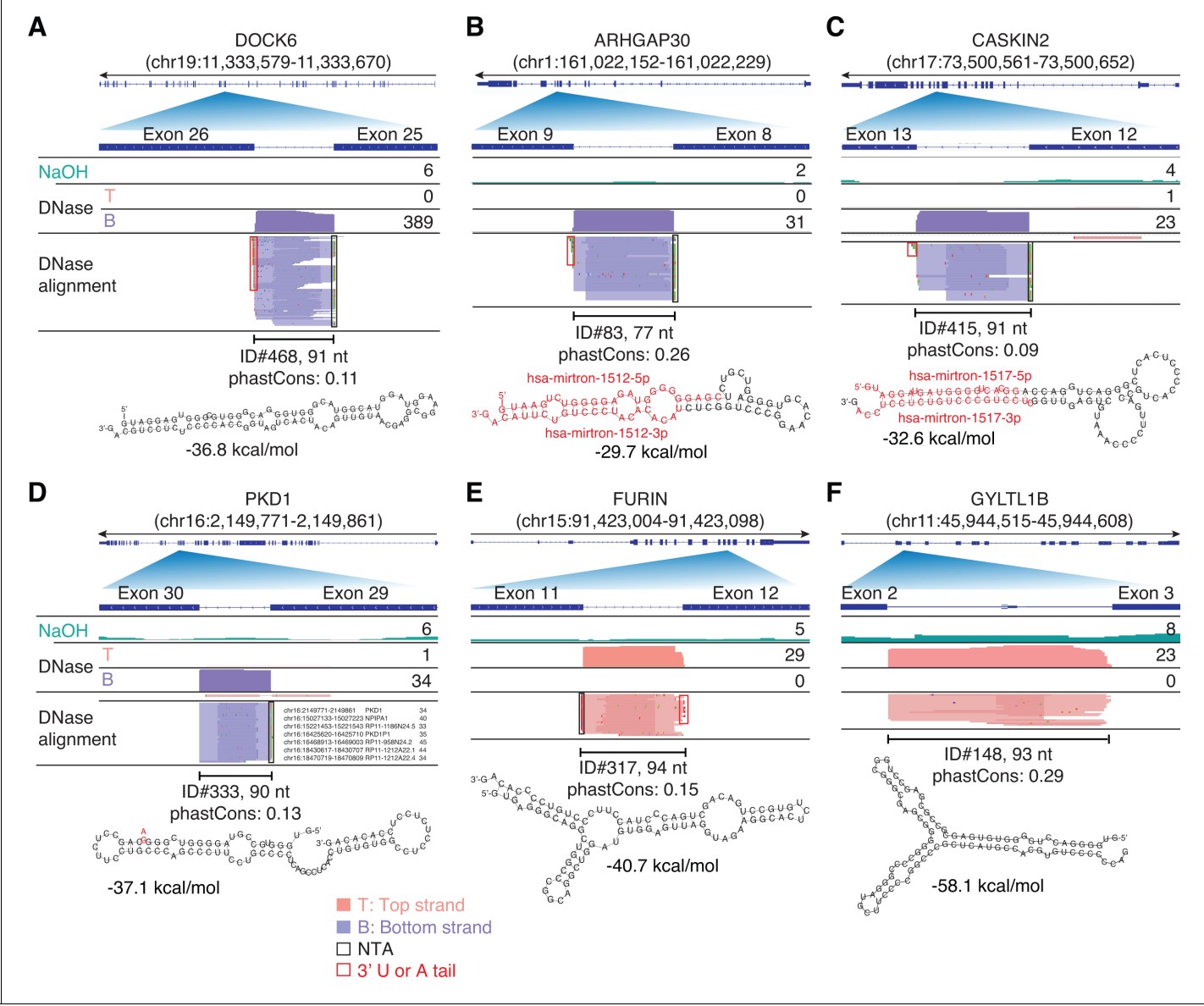

**Figure 9.** IGV screenshots showing examples of peaks corresponding to putatively structured, full-length excised intron RNAs detected in plasma by TGIRT-seq peak calling. (**A–C**) Full-length excised intron RNAs that correspond to annotated agotrons or mirtrons. (**D–F**) Full-length excised intron RNAs that do not correspond to annotated agotrons or mirtrons. Gene names are indicated at the top with the hg19 coordinates of the called peak in parentheses and an arrow below indicating the 5′ to 3′ orientation of the encoded RNA. The top track shows the gene map (exons, thick bars; intron, thin lines), with the relevant part of the gene map expanded below. The tracks below the gene map show coverage plots and read alignments for the peak region in combined datasets for NaOH-treated plasma DNA (n = 4; combined top and bottom strands, turquoise) or DNase-treated plasma RNA (n = 15; Top strand (T), pink and Bottom strand (B), purple) based on the number of deduplicated reads indicated at the right in the coverage tracks. Colors other than pink or purple in the read alignments indicate bases that do not match the reference sequence (red, green, blue, and brown indicate thymidine, adenosine, cytidine, and guanosine, respectively). The peak called by MACS2 is delineated by a bracketed line with the peak ID, length, and phastCons score for 46 vertebrates including humans indicated below (see phylogenetic tree in *Figure 10*). The most stable predicted secondary structure and its MFE (ΔG) computed by RNAfold are shown below the peak ID. The annotated mature miRNA and passenger strands of mirtrons are highlighted in red within the predicted secondary structure. Red boxes, short (1–2 nt) non-templated 3′ U or A tails; NTA/black boxes are non-templated nucleotides added by TGIRT-III to the 3′ end of cDNAs (5′ end of the RNA sequence) during TGIRT-seq library preparation.

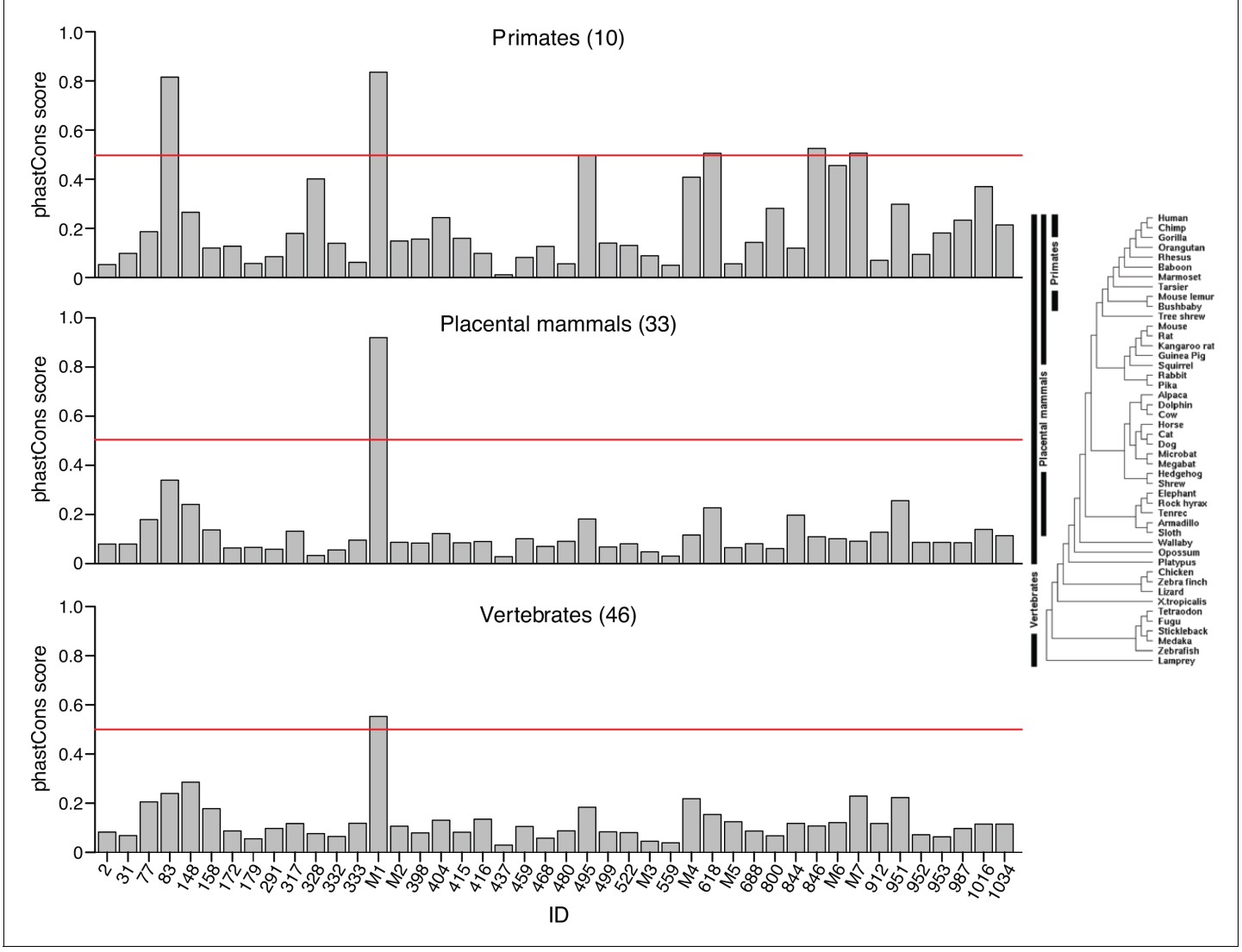

**Figure 10.** Bar graphs showing phastCons scores for 44 full-length excised introns detected in human plasma. PhastCons scores were calculated for 10 primates, 33 placental mammals, and 46 vertebrates shown in the phylogenetic tree on the right. PhastCons calculates the probability that a nucleotide belongs to a conserved sequence with the score for the peak calculated by averaging the score for each nucleotide within the peak. Scores can range from 0 to 1, with 1 being the most conserved, and 0 being the least conserved. The red line indicates a phastCons score of 0.5. M1-7 are full-length excised intron RNAs that encode non-high confidence miRNA: M1, mir-1225; M2, mir-4721; M3, mir-7108; M4, AC068946.1; M5, mir-6821; M6, mir-6891; M7, mir-1236.

denoted PKD1 in *Figure 4F*), did not correspond to an identified agotron or mirtron (Supplementary File; additional examples in *Figure 9E and F*). In initial mapping, the reads mapping to *PKD1* intron 29 were distributed between the *PKD1* gene and the same intron in six *PKD1* pseudogenes (*PKD1P1-P6*), which are located on a distant duplicated segment of the same chromosome (chromosome 16). Notably, the introns in the parent and all six pseudogenes differed by only a single nucleotide at a known SNP position and were more conserved than the flanking *PKD1* exons (6–8 mismatches in the 211 bp 5' exon and 1–3 mismatches in the 127 bp 3' exon), possibly reflecting that this intron has a critical sequence-dependent function. Nevertheless, only one of the full-length excised introns, *PKD1* intron 45 (ID# M1; encoding miR-1225), was conserved in sequence across vertebrates and only six were conserved in sequence across primates (phastCons scores $\geq$ 0.5; *Figure 10*).

## Peaks mapping within introns

Sixteen peaks identified as corresponding to annotated RBP-binding sites mapped within introns (see above), and an another 15 such peaks were found among those mapping to long RNAs but lacking an annotated RBP-binding site. Paralleling findings for mRNA fragments in plasma, most of these intron peaks (25 peaks, 81%) could be folded by RNAfold into a secondary structure with $\Delta G \leq -14.6$ kcal/mol (examples shown in *Figure 11A–E* and the remainder in *Figure 11—figure supplement 1A–D*). Several of these peaks had noteworthy features. In three cases (*Figure 11B and C* and *Figure 11—figure supplement 1A*), part of the predicted stem-loop structure for the called peak corresponded to a separate 19-nt RNA (red), which comprised a major component of the peak. In one of these peaks, the 19-nt RNA (red) was accompanied by a complementary 22-nt anti-sense RNA (blue; *Figure 11B*), and in the other two, the 19-nt RNA (red) was part of a longer 48-nt tandem repeat unit (green) within the predicted stem-loop structure (*Figure 11C* and *Figure 11—figure supplement 1A*). Two other peaks, one corresponding to an annotated binding site for the dsRNA-binding protein ILF3 (peak ID#677), contained complementary segments of long (46 and 65 bp) inverted repeats (*Figure 11D and E*). Another of the putatively structured intron RNA fragment peaks (ID#731) was identified by snoGPS (*Schattner et al., 2004*) as being able to form secondary structures resembling an H/ACA-box snoRNA, but these did not correspond to the most stable secondary structure predicted by RNAfold for this peak (*Figure 11—figure supplement 1C*).

The six intron peaks that could not be folded into stable secondary structures had other features that might contribute to nuclease resistance in plasma. Three of these peaks consisted of AG-rich sequences or tandem repeats (*Figure 11F and G*, and ID#737), including one with tandem AGAA repeats identified as an annotated binding site for TRA2A, a protein that helps regulate alternative splicing (*Figure 11G*; *Best et al., 2014*). Two others contained one arm of a long inverted repeat sequence, whose complementary arm lies outside of the called peak (ID#s 756 and 761), and the remaining peak was a highly AU-rich RNA (74% AU; ID# 964; *Supplementary file 1*).

Finally, we found that the remaining nine peaks (71–295 nt) mapping to internal segments of intron RNAs were part of retrotransposed mRNAs sequences (432–1,745 nt) that had integrated into seven different long introns (8 to 138 kb; *Figure 11H and I*, *Figure 12*, *Figure 12—figure supplement 1*). These retrotransposed mRNA sequences were identified by a BLAT search for > 95% identity to the called peak in the human genome reference sequence and originated from six different highly expressed mRNAs (ribosomal protein mRNAs *RPS3*, *RPL18A*, and *RPL41*; translation elongation factor mRNAs *EEF1A1* and *EEF1G*; and β-actin (*ACTB*) mRNA). All of these retrotransposed mRNA sequences had a short poly(A) tail (9–22 nt) and were flanked by short direct repeat target site duplications (TSDs; *Figure 12*, *Figure 12—figure supplement 1*), hallmarks of LINE-1 RT-mediated retrotransposition (*Richardson et al., 2015*). In all cases, the RNA peak identified in plasma corresponded to a smaller segment of the retrotransposed mRNA sequence that could be folded by RNAfold into a stable RNA secondary structure ($\Delta G \leq -16.4$ kcal/mole; *Figure 11H and I*, *Figure 11—figure supplement 1D*).

Comparison of genomic sequences showed the retrotransposition events that inserted the mRNA sequences within the introns were relatively recent, with six occurring in primates and two in placental mammals (*Figure 12A* and *Figure 12—figure supplement 1*). In all of these cases (and three additional cases of retrotransposed mRNA sequences described below), the RNA peak identified in plasma corresponded to a segment of the retrotransposed mRNA whose genomic sequence was identical to that in the gene encoding the functional mRNA (verified by searching the peak sequence against the hg19 human reference genome using BLAT and manually checking the sequences in the human genome browser and IGV), making it impossible to determine the origin of the RNA detected in plasma. Analysis of two introns with the largest number of homologous genomic sequences across primates and placental mammals showed that both the retrotransposed mRNA sequence and called peak region were more conserved than the flanking intron sequences, consistent with functional importance (*Figure 12B*).

## Peaks corresponding to unannotated genomic loci

Only three of the peaks identified by peak calling mapped to unannotated genomic loci (*Figure 4A*). One corresponded to a different putatively structured 3' segment of a retrotransposed *ACTB* mRNA, with a short poly(A) tail flanked by two different Alu elements with a third Alu element

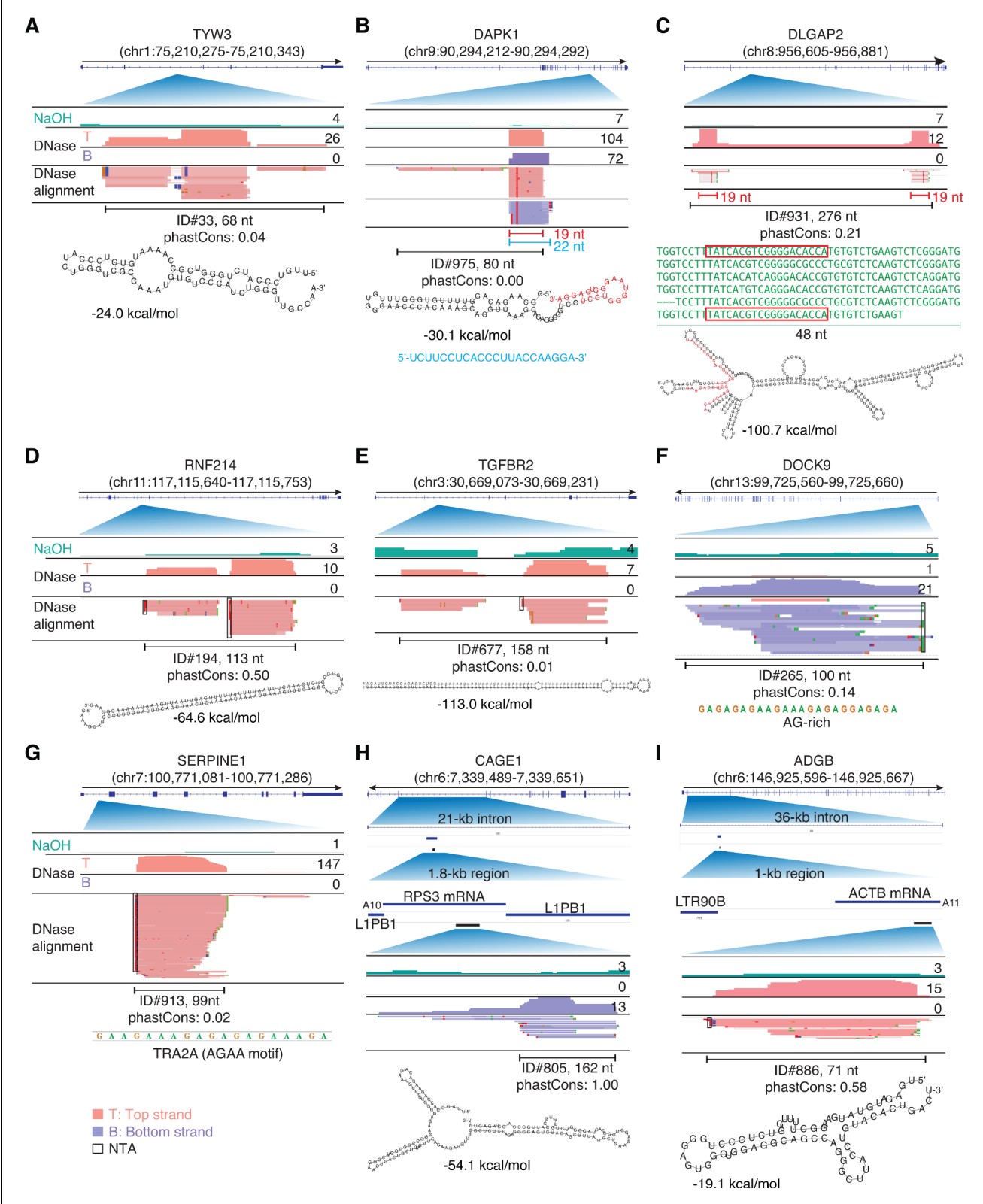

**Figure 11.** IGV screenshots showing examples of peaks mapping within introns. Gene names are indicated at the top with the hg19 coordinates of the called peak in parentheses and an arrow below indicating the 5' to 3' orientation of the encoded RNA. IGV screenshots are labeled as in *Figure 9*. (A– E) Peaks corresponding to intron RNA fragments that can be folded into stable stem-loop structures. The predicted stem-loop structure for the peak in panel B includes a 19-nt segment (red), which corresponds to a separate 19-nt RNA that is a major component of the called peak and was

*Figure 11 continued on next page*

*Figure 11 continued*

accompanied by a separate 22-nt antisense RNA (blue) that is complementary to the 19-nt RNA. The predicted stem-loop structure in panel C is comprised of imperfect tandem repeats of a 48-nt sequence (green), part of which (red) corresponds to a separate 19-nt RNA that maps with fewest mismatches to the two terminal repeat units. The peaks in panels D and E are comprised of two complementary segments of long inverted repeats (46 bp with no mismatches and 65 bp with one mismatch, respectively), with the peak in panel E corresponding to an annotated binding site for double-stranded RNA-binding protein ILF3. (F, G) Peaks corresponding to intron RNA fragments that are not predicted by RNAfold to fold into stable secondary structures. (H, I). Peaks corresponding to short, putatively structured segments of mRNAs that had retrotransposed into long introns. The top track shows the gene map with the long intron to which the peaks mapped expanded below. The small blue line delineates the retrotransposed mRNA sequences with a short poly(A) tail and proximate LINE-1 elements expanded below. The small black bar delineates the region containing the called peak that is expanded in the IGV plots below. The most stable predicted secondary structure and its MFE (ΔG) computed by RNAfold are shown below the peak ID. When necessary, reads shown in alignment tracks were down sampled to a maximum of 100 for display. NTA/black boxes are non-templated nucleotides added during TGIRT-seq library construction.

The online version of this article includes the following figure supplement(s) for figure 11:

**Figure supplement 1.** IGV screenshots and predicted secondary structures for peaks corresponding to intron RNA fragments.

inserted within the retrotransposed mRNA sequence (*Figure 4—figure supplement 3A*). The second was a 72-nt peak that included multiple copies of a 17-nt RNA with discrete 5'- and 3'-ends (*Figure 4—figure supplement 3B*), and the third was a 158-nt peak consisting of TCCAT(C)$_{4-6}$GTG repeats (*Figure 4—figure supplement 3C*).

### Peaks corresponding to antisense RNAs

Finally, 25 peaks were identified as antisense transcripts of annotated genomic features (*Figure 4A*). Eight of these peaks mapped to introns and the remaining 17 peaks mapped to mRNA exons or pseudogenes. Among the eight antisense peaks mapping to introns, four corresponded to or contained multiple reads for four different discrete short RNAs (< 20 nt) complementary to a sequence within the intron (*Figure 13A–D* and *Supplementary file 1*), with one corresponding to a 13-nt segment of miR-4497, a non-high-confidence predicted miRNA encoded at another locus (*Figure 13A*), and another corresponding to a 17-nt segment of a 36-nt tandem repeat (*Figure 13C*). Two other antisense peaks mapping to introns were putatively structured segments (ΔG ≤ −22 kcal/mol) of retrotransposed *TMSB4X* and *FTH1* mRNAs that had been inserted within introns in the antisense orientation relative to the host gene (*Figure 13—figure supplement 1A and B*), and the remaining two peaks (ID#973 and 539) contained RNA fragments with heterogenous 5' or 3' ends, with one (ID#539) having the potential to fold into a stable secondary structure (ΔG = −26.7 kcal/mol; *Figure 13—figure supplement 1C and D*).

Among the 17 other antisense peaks mapping to mRNAs or pseudogenes, 12 mapped to hemoglobin genes (*HBA2*, *HBA1*, *HBQ1*, *HBB*, etc.), four to other highly represented genes (*RPS29*, *TMSB4X*, *INPP5E*, and *FTL*), and one to a pseudogene (*Figure 13E–H* and *Figure 13—figure supplement 1*). The peak mapping to *INPP5E* included a discrete 17-nt RNA (*Figure 13E*). While many of these peaks may correspond to *bona fide* antisense transcripts, some included antisense reads that extended across two or more spliced exons (horizontal orange boxes) and/or were mirrored by a partially overlapping antisense DNA peak (red boxes extending vertically across the coverage tracks and read alignments; *Figure 13F–H* and *Figure 13—figure supplement 1*), the latter suggesting that they contain DNA fragments that are partially protected from DNase I-digestion in an RNA/DNA duplex. Such RNA/DNA duplexes could be remnants of R-loops formed during transcription (*Sanz et al., 2016*) or cDNAs generated by reverse transcription of a spliced mRNA by an endogenous cellular RT. None of the Reads 1 of these possible antisense DNAs began with an R2 adapter sequence as would be expected for recopying of an initial cDNA by TGIRT-III.

### Discussion

Here, we used TGIRT-seq combined with peak calling to comprehensively profile RNAs present in apheresis-prepared human plasma pooled from multiple healthy individuals. Previous TGIRT-seq analysis of plasma RNAs from a healthy male individual showed that human plasma contains largely fragmented mRNAs originating from > 19,000 different protein-coding genes, together with abundant full-length mature tRNAs and other structured sncRNAs (*Qin et al., 2016*). Here, by using an

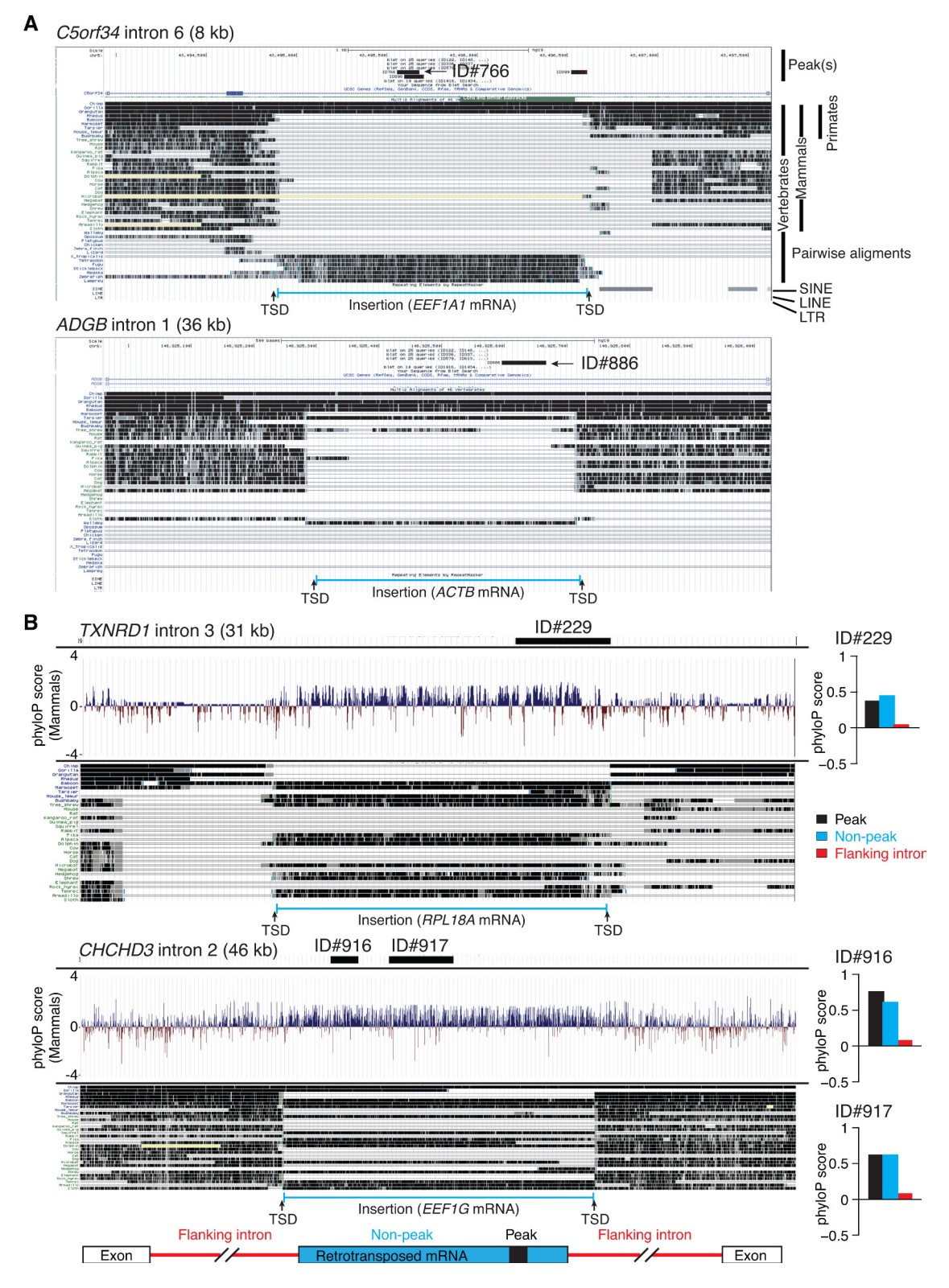

**Figure 12.** Genome browser (UCSC) screenshots showing examples of long introns containing retrotransposed mRNAs sequences that are called as peaks and sequence conservation across such introns. (**A**) Two examples of long introns containing retrotransposed mRNA sequences that are called as peaks. The host gene, intron number, and intron length are indicated at the top, with the called peak or peaks (thick black line(s)) and peak ID indicated in the top track. The tracks below show alignments of genomic sequences of different organisms. Taxonomic ranks are shown at the top right, *Figure 12 continued on next page*

*Figure 12 continued*

and the complete phylogenetic tree for the aligned sequences is shown in (*Figure 12—figure supplement 1*). The locations of nearby SINE, LINE, and LTR-containing retroelements in the human genome are shown below the alignments. The inserted retrotransposed mRNA sequence (bracketed blue line at the bottom) is identified by gap in genomic sequence for organisms that do not contain the insertion and by target site duplications (TSDs; 5–20 nt direct repeats) flanking the inserted sequence. Additional examples of long introns containing retrotransposed mRNA sequences that are called as peaks are shown in *Figure 12—figure supplement 1*. (B) Sequence conservation as measured by phyloP scores across long introns containing retrotransposed mRNAs sequences that are called as peaks. The gene name, intron number, and length of the intron are indicated at the top, with the position of the called peak delineated by a thick black line. The track below shows phyloP scores for placental mammals based on the pairwise alignment of genomic sequences from the different organisms in the genome browser alignments below. The inserted retrotransposed mRNA sequence (bracketed blue line at bottom) is identified by a gap in the genomic sequence for organisms that do not contain the insertion and by target site duplications (TSDs, 5–20 nt direct repeats) flanking the inserted sequence. The bar graphs to the right of the phyloP tracks show average phyloP scores for the called peak, flanking regions of the retrotransposed mRNA sequence, and 5'- and 3'-flanking regions of the host intron (see schematic at bottom). PhyloP scores are $-\log_{10}$ (p values) for the difference between the measured evolutionary conservation at a position in an alignment relative to that expected for a null hypothesis of neutral evolution. The score for each region was calculated by averaging the scores for each position in that region. Positive and negative scores indicate slower and faster evolution rates than expected from a neutral evolution model calculated from the phylogenetic tree (*Ramani et al., 2018*).

The online version of this article includes the following figure supplement(s) for figure 12:

**Figure supplement 1.** Genome browser (UCSC) screenshots showing additional examples of long introns containing retrotransposed mRNAs sequences that are called as peaks.

improved TGIRT-seq method with a UMI to deconvolute duplicate reads, we obtained a more quantitative view of the relationship between different classes of plasma RNAs, extended analysis to the plasma microbiome, compared mRNA sequences detected in the same plasma by TGIRT-seq and SMART-Seq, and introduced the use of a peak-calling algorithm for the analysis of TGIRT-seq datasets. The latter revealed that many of the discrete mRNA and intron RNA peaks that persist in plasma correspond to annotated RBP-binding sites and/or are predicted to have stable RNA secondary structures that may provide protection from plasma nucleases. Peak calling also provided new insights into repeat RNAs and showed that they are over-represented in called peaks consisting of uniformly sized RNA fragments. Additionally, we identified a number of unannotated miRNA-sized RNAs, some apparently processed from stem-loop structures, potential RNA/DNA hybrids apparently containing residual DNA that escapes extensive DNase digestion, and novel families of structured intron RNAs and intron RNA fragments that had not been seen previously using other RNA-seq methods. These include a family of short putatively structured full-length excised introns RNAs, subsets of which correspond to agotrons and/or mirtrons, as well as putatively structured intron RNA fragments, including a family mapping to conserved structured segments of retrotransposed mRNAs that had inserted within long introns. Although not originally a focus of our study, we also found that commercial plasma obtained by apheresis from healthy individuals had no detectable reads for pre-screened pathogens (hepatitis B virus, hepatitis C virus, HIV-1, HIV-2, and syphilis), but did contain low levels of reads for bacterial RNAs and other RNA and DNA viruses (*Figure 2*), with possible implications for plasma used for clinical purposes (*Duan et al., 2020*).

TGIRT-seq with RNA-seq adapters containing a UMI to deconvolute duplicate reads confirmed that full-length mature tRNAs are the most abundant class of sncRNAs in human plasma, followed by Y RNAs and 7SL RNA, with tRNA halves, short tRFs, and mature and pre-miRNAs and passenger strands present in lower concentrations. The full-length, mature tRNAs and other sncRNAs present in plasma may be protected from plasma nucleases by stable RNA secondary or tertiary structure, bound proteins, or encapsulation in EVs, the latter suggested by TGIRT-seq analysis of RNAs present in highly purified EVs and exosomes secreted by cultured human cells, which likewise identified full-length mature tRNAs and Y RNAs as the most abundant RNA species (*Shurtleff et al., 2017*). Although the updated TGIRT-seq methods used in this work improved the detection of miRNAs and 5'- and 3'-tRFs, further analysis of these RNAs by RT-qPCR and FirePlex assays indicated that some miRNAs remained substantially underrepresented relative to RNAs >60 nt (manuscript in preparation). Thus, while comprehensive TGIRT-seq of heterogeneously sized RNA preparations can accurately determine the relative abundance of RNAs > 60 nt (*Nottingham et al., 2016*; *Boivin et al., 2018*), determining the quantitative relationship between very short RNAs and RNAs > 60 nt requires orthogonal methods. Surprisingly, despite the size bias against very short RNAs, we

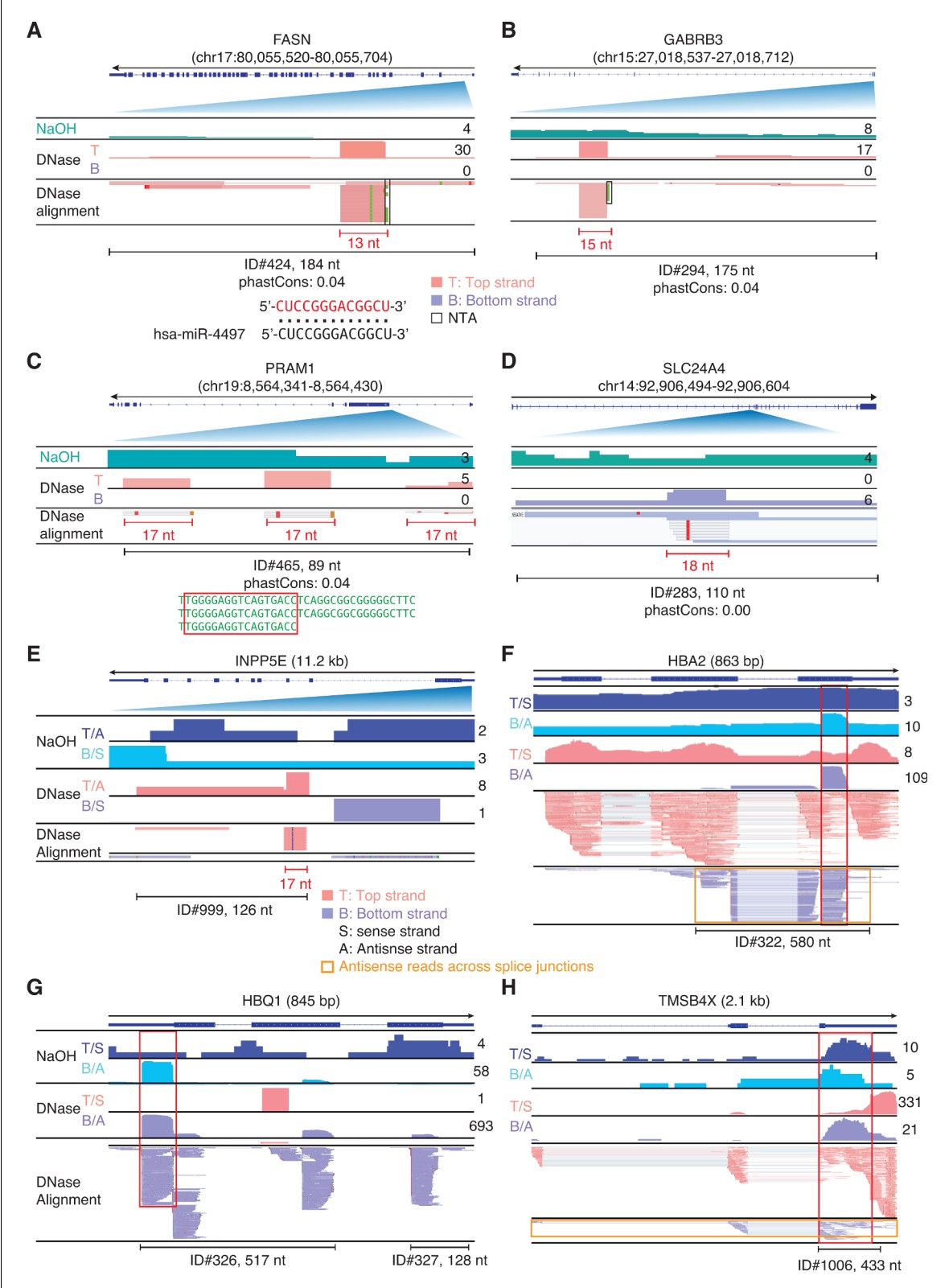

**Figure 13.** IGV screenshots showing examples of peaks corresponding to antisense RNAs detected in plasma by TGIRT-seq peak calling. Gene names are indicated at the top with the hg19 coordinates of the called peak indicated in parentheses and an arrow below indicating the 5′ to 3′ orientation of the encoded mRNA. (**A–D**) Antisense peaks that map within introns and correspond to or contain short RNAs with discrete 5′ and/or 3′ ends (red). The IGV screenshots in these panels are labeled as in *Figure 9*. The 13-nt RNA in panel A corresponds to part of miR-4497 (alignment shown below). (**E–H**)

*Figure 13 continued on next page*

*Figure 13 continued*

Antisense peaks mapping to genes that are highly represented in plasma. In these panels, separate coverage plots are shown for the top (T) and bottom (B) strands for the NaOH-treated plasma DNA samples, and top and bottom strands are additionally labeled as sense (S) or antisense (A) depending upon the 5' to 3' orientation of the mRNA encoded by the gene displayed in IGV. The red boxes extending vertically across the read alignments and coverage plots delineate RNA peaks that are mirrored by DNA peaks in NaOH-treated plasma DNA, and the horizontal orange boxes delineate antisense reads that extend across multiple spliced exons. When necessary, reads shown in alignment tracks were down sampled to maximum of 100 for display.

The online version of this article includes the following figure supplement(s) for figure 13:

**Figure supplement 1.** IGV screenshots showing additional examples of peaks mapping to the antisense strand of introns in protein-coding genes.
**Figure supplement 2.** IGV screenshots of other peaks mapping to the antisense strand of annotated genes or pseudogenes.

identified 15 novel discrete short RNAs (13–25 nt; *Figure 11*, *Figure 13*, *Figure 4—figure supplement 3*, *Figure 7—figure supplement 3*, *Figure 11—figure supplement 1*, and *Supplementary file 1*). Only one of these short RNAs (non-high-confidence miR-4497; *Figure 13A*) could be found in miRBase, piRNAdb, tRFdb or cross-linked regions in AGO1-4 PAR-CLIP datasets (*Hafner et al., 2010*), raising possibility that they represent other classes of short regulatory RNAs, whose protein-partners and functions, if any, remain to be determined.

The proportion of reads corresponding to mRNAs in plasma prepared by apheresis was relatively low (< 5% of mapped reads), but nevertheless sufficient to identify mRNA fragments originating from > 19,000 different protein-coding genes, with relatively uniform coverage across mRNA exons for highly represented genes (*Figure 3*). The number of different mRNAs detected in plasma by TGIRT-seq was comparable to that detected in parallel assays of the same plasma by ultra low input SMART-Seq v4, which uses oligo(dT) priming, and higher than that in other studies using different methods that sequence mRNA fragments with or without size selection (*Pan et al., 2017*; *Tsang et al., 2017*; *Akat et al., 2019*; *Giraldez et al., 2019*). Notably, the size distribution of cDNAs generated by ultra low input SMART-Seq suggested that a substantial proportion of the mRNA fragments present in plasma have a length > 200 nt (36%; *Figure 3D*), which would be lost by incorporating a size-selection step. As TGIRT-seq is a comprehensive RNA-seq method, the presence of other more abundant RNA biotypes is an inherent limitation for mRNA detection. However, the proportion of TGIRT-seq reads corresponding to plasma mRNAs could be increased by depleting rRNAs, by using a maximally efficient RNA extraction kit, and possibly by using plasma prepared from freshly drawn blood to minimize degradation of more labile mRNA fragments, as was done in the previous TGIRT-seq analysis of human plasma RNAs, which gave substantially higher proportions of mRNA reads (*Qin et al., 2016*).

The mRNA fragments present in plasma are likely a mixture of transient RNA degradation intermediates and more persistent RNA fragments that are protected from plasma nucleases by bound proteins, RNA structure, encapsulation in EVs, or some combination of these factors. Both the present and previous TGIRT-seq analyses indicated that many of the mRNA present in plasma have 3' OHs, which are required for efficient 3' RNA-seq adapter addition by TGIRT-template-switching, particularly under the high-salt reaction conditions used in the previous study (*Qin et al., 2016*). As most cellular enzymes that function in RNA processing or turnover leave 3' OH groups while extracellular RNases, such as RNase 1, leave 3' phosphates (*Houseley and Tollervey, 2009*; *Sorrentino, 2010*; *Schoenberg and Maquat, 2012*), these findings suggest that a high proportion of the mRNA fragments detected in plasma were generated by intracellular RNases that function in mRNA turnover rather than degradation of intact mRNAs that were released into plasma.

To identify those mRNA fragments in plasma that might be associated with bound proteins, we used a peak-calling algorithm and intersected the called RNA peaks with ENCODE eCLIP annotated RBP-binding site sequences. We thus identified numerous discrete mRNA and lncRNA fragments that are annotated binding sites for ~100 different RBPs, most of which were previously identified as part of the plasma proteome. Notably, we found plasma RNAs are particularly enriched in binding sites for stress granule proteins (including DDX3X, DDX6, DHX30, LARP4, IGF2BP3, PABPC4, PCBP1, RPS3, SERBP1, SND1, and YBX3; *Figure 6* and *Figure 6—figure supplement 1*), consistent with a previously suggested link between RNP granules and RNA loading into secreted extracellular vesicles, a vehicle by which some mRNA fragments may enter plasma (*Shurtleff et al., 2017*; *Temoche-Diaz et al., 2019*). Such a link could also explain the prevalence of 5' TOP mRNAs, which

are translationally repressed by recruitment to RNP granules under stress conditions, in both plasma and EVs secreted by cultured human cells (*Damgaard and Lykke-Andersen, 2011*; *Shurtleff et al., 2017*). Although our findings do not directly demonstrate the association of the annotated RBP with the protected RNA fragment, collectively they suggest that many of the mRNA fragments present in plasma are protected from plasma nucleases by bound proteins, similar to the inferred protection of plasma DNA fragments by nucleosomes packaging or by the binding of transcription factors or other proteins (*Chandrananda et al., 2015*; *Snyder et al., 2016*; *Wu and Lambowitz, 2017*). The number of protein-protected mRNA fragments that we identified in plasma is likely underestimated as our search was restricted to ENCODE eCLIP annotated RBP-binding sites and used plasma prepared with the anti-coagulant EDTA, which destabilizes some RNPs (*Dhabi et al., 2013*).

In addition to bound proteins, many of the mRNA fragments that we detected in plasma with or without annotated RBP-binding site sequences were predicted by RNAfold to form stable RNA secondary structures that might contribute to their protection from plasma nucleases. The predicted RNA secondary structures of some mRNA fragments could be classified by Infernal/Rfam as corresponding to known structural elements (*e.g.*, histone 3' stem-loop structures, iron response elements, selenocysteine insertion sequence, and pre-miRNA stem-loop structure; *Figure 7*), but in most cases, the predicted secondary structure could not be classified by Infernal/Rfam, and its functional significance, if any, remains to be evaluated. Some peaks corresponding to structured mRNA fragments had discrete 5' and 3' ends and could be unannotated sncRNAs (*Figure 7—figure supplement 3A–D*), while others consisted of or included discrete miRNA-sized RNA, possibly unannotated short regulatory RNAs (*Figure 7—figure supplement 3E–G*).

A surprisingly high proportion of the discrete RNAs identified by peak calling (23.1%) corresponded to Repbase-annotated short tandem repeat and transposable element RNAs (*Figure 4B*). TGIRT enzymes have been found to more completely reverse transcribe and better quantitate short tandem repeat RNAs than do retroviral RTs and can also give full-length reads of Alu and other structured SINE element RNAs (*Katibah et al., 2014*; *Carrell et al., 2018*). In all, TGIRT-seq detected discrete RNAs containing 37 different types of short tandem repeats, including telomeric and centromeric repeats. The short tandem repeat RNAs had a narrow size distribution (peak at ~80 nt) with most encompassing a high proportion of the genomic repeat unit (*Figure 8*), possibly reflecting protection of the repeat RNA from plasma nucleases by bound proteins or RNA structural features. Many but not all of the transposable element and endogenous retroviral RNAs were RNA fragments of a size similar to that of the simple repeat RNAs. The ability of TGIRT-seq to profile repeat RNAs in plasma may be useful for liquid biopsy of RNA repeat expansion diseases, such as myotonic dystrophy and some forms of amyotrophic lateral sclerosis.

TGIRT-seq peak calling also identified novel intron RNAs, including a family of short, full-length excised intron RNAs that had not been seen previously using other RNA-seq methods. These full-length excised introns ranged in length from 73 to 130 nt, are likely linear RNAs, had RNAfold-predicted secondary structures with $\Delta Gs \leq -18.7$ kcal/mole, and in many cases, corresponded to annotated binding sites of spliceosomal proteins and/or AGO1-4. Twenty three of the 44 full-length excised intron RNAs that we detected in plasma corresponded either to annotated or potential mirtrons, intron pre-miRNAs that are processed by DICER to produce mature miRNAs (*Berezikov et al., 2007*; *Okamura et al., 2007*; *Ruby et al., 2007*) and/or to annotated or potential agotrons, structured excised intron RNAs that bind AGO proteins and function directly to repress mRNA translation (*Hansen et al., 2016*; *Hansen, 2018*). Seven of the full-length excised intron RNAs that we detected in plasma were annotated as both an agotron and mirtron, blurring the distinction between these two classes of RNAs. Notably, although we detected mirtron pre-miRNAs in plasma, we did not detect the corresponding mature miRNAs, raising the possibility that mirtron pre-miRNAs are selectively exported or have higher stability in plasma. While extracellular miRNAs have been suggested to function in intercellular communication (*Valadi et al., 2007*; *Lin et al., 2020*), mirtron pre-miRNAs and agotrons may be as well or better suited for this role, with the mirtron pre-miRNAs entering cells as precursor RNAs that could be processed by DICER into mature miRNAs, and agotrons bound to AGO proteins entering cells as an RNP that could function directly in regulating gene expression in a miRNA-like manner.

Surprisingly, almost half (21 of 44) of the structured full-length excised introns that we detected in plasma did not correspond to identified agotrons or mirtrons. These introns could be mirtrons or agotrons that have yet to be annotated, or they could have some other function that remains to be

determined. One of the most abundant full-length excised intron RNAs that we detected in plasma, *PKD1* intron 29 (ID#333; *Figure 9*), was highly conserved in both the parent gene and six distant pseudogenes (one mismatch at a known SNP position), possibly reflecting selection for a sequence-dependent function of this intron. Nevertheless, only one of the full-length excised intron RNAs that we detected in plasma, a putative mirtron encoding miR-1225, was conserved in sequence across vertebrates and mammals, and only six (not including *PKD1* intron 29) were conserved in sequence across primates (phastCons score ≥ 0.5; *Figure 10*), suggesting that any sequence-dependent function of these intron RNAs would have been recently acquired.

In addition to full-length excised introns, peak calling identified a variety of intron RNA fragments, with more than half (16 of 31) corresponding to annotated RBP-binding sites and most having stable predicted RNA secondary structures or other structural features (tandem repeats, inverted repeats, or highly AG- or AU-rich sequences) that may afford protection from plasma nucleases. The RBPs potentially associated with these intron RNA fragments were a largely different set than those associated with the full-length excised intron RNAs and in some cases were related to a specific RNA splicing function or structural feature of the intron. Thus, an intron RNA peak containing complementary segments of a long-inverted repeat corresponded to an annotated binding for the dsRNA-binding protein ILF3, and an intron RNA peak containing tandem AGAA repeats corresponded to annotated binding site for the alternative splicing factor TRA2A, which recognizes this motif (*Figure 11*). Some of the intron RNA peaks that could be folded into stem-loop structures also contained unannotated discrete miRNA-sized RNAs (13–25 nt), which may have been processed out of the stem-loop structure. The latter finding raises the possibility that whatever their origin or original function, if any, structured excised intron RNAs in human cells may be accessible to DICER or other RNase III-like enzymes that cleave double-strand RNAs into discrete small fragments to generate a repository of non-coding short RNAs that could evolve to have miRNA-like functions.

By analyzing RNA peaks that mapped to both introns and mRNAs, we also identified multiple human introns containing retrotransposed segments of highly expressed mRNAs, conserved, structured portions of which persist in plasma (*Figure 11*, *Figure 11—figure supplement 1D*, *Figure 12*, *Figure 12—figure supplement 1*). In all cases, the retrotransposition event that inserted the mRNA sequence within the intron appears to have been mediated by a LINE 1 RT and was relatively recent, with most occurring in primates and a few occurring earlier in placental mammals. In all cases, the RNA peak detected in plasma matched a putatively structured portion of the mRNA sequence that was highly conserved within the intron, raising the possibility of functional importance (*Figure 12B*). Additionally, for several highly expressed mRNAs, including three that were progenitors of processed pseudogenes in the human genome (*TMSBX4*, *RPS29*, and *FTL*), we detected antisense reads that could be cDNAs generated by an endogenous RT, possibly precursors of continuing retrotransposition events in human cells (*Figure 13* and *Figure 13—figure supplement 1*).

The ability of TGIRT-seq to simultaneously profile a wide variety of RNA biotypes in human plasma, including structured RNAs that are intractable to retroviral RTs, may be advantageous for identifying optimal combinations of coding and non-coding RNA biomarkers that could then be incorporated in target RNA panels for diagnosis and routine monitoring of disease progression and response to treatment. The finding that some mRNAs fragments persist in discrete called peaks suggests a strategy for identifying relatively stable mRNA regions that may be more reliably detected than other more labile regions in targeted liquid biopsies. Finally, we note that in addition to their biological and evolutionary interest, short full-length excised intron RNAs and intron RNA fragments, such as those identified here, may be uniquely well suited to serve as stable RNA biomarkers, whose expression is linked to that of numerous protein-coding genes.

## Materials and methods

**Key resources table**

| Reagent type (species) or resource | Designation | Source or reference | Identifiers | Additional information |
|---|---|---|---|---|
| Biological sample (*Homo sapiens*) | Plasma | Innovative Research | IPLA-N-K2E | Human plasma pooled from healthy individuals |

*Continued on next page*

*Continued*

| Reagent type (species) or resource | Designation | Source or reference | Identifiers | Additional information |
|---|---|---|---|---|
| Commercial assay or kit | QIAamp ccfDNA/RNA kit | Qiagen | Qiagen 55184 | |
| Commercial assay or kit | SMART-Seq v4 Ultra Low Input RNA kit | Takara Bio | R400752 | |
| Commercial assay or kit | TGIRT-III reverse transcriptase | InGex, LLC | TGIRT | |

## Preparation of RNA and DNA from human plasma

Human plasma pooled from healthy individuals (IPLA-N-K2E) was purchased from Innovative Research. The plasma was prepared by apheresis into K2-EDTA tubes and was certified by the provider as testing negative for HBV, HCV, HIV-1, HIV-2, and syphilis using FDA-approved methods.

Nucleic acids were isolated from 4 mL of plasma by using a QIAamp ccfDNA/RNA kit (Qiagen 55184) following the manufacturer's protocol, with the nucleic acids eluted in a final volume of 20 µL. To obtain plasma DNA, RNA was degraded by adding 1 µL of 5 N NaOH and incubating at 95°C for 3 min. The solution was then neutralized by adding 1 µL of 5 N HCl, and the products were cleaned up with an Oligo Clean and Concentrator kit (Zymo Research), with the DNA eluted in a final volume 10 µL.

To obtain plasma RNA, DNA in the initial 20 µL nucleic acids solution was digested by adding 16 µL DNase I (1 U/µL) plus 4 µL 10X DNase I buffer (DNase I set; Zymo Research), incubating for 15 min at room temperature, and cleaning up with a Zymo RNA Clean and Concentrator kit, with the RNA eluted in a volume of 10 µL. For one sample (ExoI_1), the DNase I-treated plasma RNA (~2 ng in 10 µL of eluant) was denatured by heating to 95°C for 3 min and immediately placed on ice, then digested with exonuclease I by adding 1 µL of enzyme (20 U/µL) plus 1.5 µL 10X Reaction Buffer (Lucigen) and incubating 1 hr at 37°C. The products were then cleaned up with a Zymo RNA Clean and Concentrator kit eluted in a final volume of 10 µL. In two other samples, the initial 20 µL nucleic acids sample was split into 10 µL aliquots and digested with Exonuclease I as above. One aliquot was cleaned up with a Zymo RNA Clean and Concentrator kit (ExoI_3), and other was treated with DNase I as above prior to clean up (ExoI_2; *Supplementary file 1*).

To remove 3' phosphates and 2', 3' cyclic monophosphates, 2.5 µL of T4 polynucleotide kinase (5 U/µL) plus 2.5 µL 10X Reaction Buffer (Lucigen) were added to the initial 20 µL nucleic acids solution, and the samples were incubated for 30 min at 37°C, followed by incubation for 5 min at 70°C to denature the enzyme and clean up with a Zymo RNA Clean and Concentrator eluted in a volume of 20 µL. The samples were then incubated with DNase I prior to final cleaned up as above.

Fragmentation of plasma RNA was done by using an NEBNext Magnesium RNA Fragmentation Module (New England Biolabs) following the manufacturer's protocol. 2 µL of 10X Fragmentation Buffer was added to 20 µL of nucleic acids and incubated for 2 min at 94°C. The fragmented RNA was then cleaned up with a Zymo RNA Clean and Concentrator kit, treated with T4 polynucleotide kinase to remove 3' phosphates and 2', 3' cyclic phosphates, cleaned up again with an RNA Clean and Concentrator kit, and treated with DNase I prior to final clean up as above.

## TGIRT-seq library construction

TGIRT-seq library construction was done as described (*Xu et al., 2019*), with modified RNA-seq adapters that decrease adapter-dimer formation and add a 6-nt unique molecular identified (UMI; six randomized nucleotides) in reaction medium containing 200 mM NaCl, which increases the efficiency of 3'-RNA-seq adapter addition by TGIRT template-switching (*Lentzsch et al., 2019*). For the initial template-switching and reverse transcription reactions, 10 µL of plasma RNA or DNA was pre-incubated for 30 min at room temperature with 1 µM TGIRT-III (InGex, LLC) and 100 nM of synthetic RNA template/DNA primer substrate in reaction medium containing 200 mM NaCl, 5 mM MgCl$_2$, 20 mM Tris-HCl, pH 7.5, and 5 mM dithiothreitol. The synthetic RNA template/DNA primer substrate consisted of a 35-nt RNA oligonucleotide that contains an Illumina read 2 (R2) sequence with a 3'-blocking group (C3 spacer, 3SpC3; IDT) annealed to a complementary 36-nt DNA primer that leaves a single nucleotide 3'-DNA overhang (an equimolar mixture of A, C, G, and T residues), which directs

TGIRT-template switching by base pairing to the 3′ nucleotide of a target RNA. Reactions were initiated by adding 1 mM dNTPs (an equimolar mix of 1 mM each dATP, dCTP, dGTP, and dTTP), incubated for 15 min at 60°C, and terminated by adding 0.25 M NaOH and incubating at 95°C for 3 min, followed by neutralization with 0.25 M HCl, with the resulting cDNAs cleaned up by using a MinElute Reaction Cleanup Kit (Qiagen). The second adapter, a 5′-adenylated/3′-blocked (C3 spacer, 3SpC3; IDT) DNA containing the reverse complement of an Illumina read one sequence (R1R) and the 6-nt UMI at its 5′ end was then ligated to the 3′ end of the cDNAs by using Thermostable 5′ AppDNA/ RNA Ligase (New England Biolabs), as described (*Wu and Lambowitz, 2017*). After clean up with a MinElute Reaction Cleanup Kit (Qiagen), ligation products were amplified by PCR using a KAPA Library Amplification Kit (KAPA Biosystems KK2610) with 500 nM of Illumina multiplex primer (a 5′ primer that adds a P5 capture site) and 500 nM of index primer (a 3′ primer that adds an index and a P7 capture site). PCR was done with initial denaturation at 98°C for 30 s followed by 12 cycles of 98°C for 45 s, 60°C for 15 s, and 72°C for 30 s, and a final incubation at 72°C for 5 min. The PCR products were purified by using 1.3X Agencourt AMPure XP beads (2 or 3 cycles; Beckman Coulter) and evaluated by using an Agilent High Sensitivity DNA Kit on an Agilent 2100 Bioanalyzer (Agilent). TGIRT-seq libraries were sequenced on an Illumina NextSeq 500 instrument (75-nt paired-end reads) at the Genome Sequencing and Analysis Facility (GSAF) at the University of Texas at Austin.

## SMART-Seq library preparation

20 µL of nucleic acid sample from 4 mL of plasma was treated with DNase I as described above, and the resulting DNase I-treated RNA (~2 ng) was processed for sequencing by using a SMART-Seq v4 Ultra Low Input RNA kit (Takara Bio) according to the manufacturer's protocol with 12 cycles of PCR followed by two rounds of clean up with 1.3X Agencourt AMPure XP beads (Beckman Coulter). The resulting double-stranded DNAs were fragmented with a Covaris sonicator (S220, Woburn, MA) using a 10% duty cycle, intensity at 5, and 200 cycles per burst, and Illumina-sequencer-compatible libraries were constructed by using an NEBNext Ultra II DNA Library Prep Kit (New England Biolabs).

## Sequence data processing

FASTQ files were processed by using the TGIRT-map pipeline (*Wu et al., 2018*) with modifications for preserving UMI information and more stringent read trimming and filtering, including trimming of partial adapters and trimming each adapter sequence twice for each read. 6-nt UMIs were clipped and appended to the read ID, and only read pairs with an average UMI PHRED quality score $\geq$ 20 were retained. Adapters, sequencing artifacts (*e.g.*, long homopolymers), and TGIRT-seq byproducts (*e.g.*, primer dimers) were removed with Atropos (*Didion et al., 2017*) using options: trim -U 1 –minimum-length=15 –threads=24 –no-cache-adapters –overlap 3 –nextseq-trim=25 –times=2 –max-n=3 –error-rate=0.2 –front AAGATCGGAAGAGCACACGTCTGAACTCCAGTCACNNNNNNNATCTCGTA TGCCGTCTTCTGCTTG –anywhere=GCTCTTCCGATCTT -b GCACACGTCTGAACTCCAGTCAC -b GTGACTGGAGTTCAGACGTGTGCTCTTCCGATCTT -a A{100} -a T{100} -G GTGACTGGAGTTCA-GACGTGTGCTCTTCCGATCTT -B AAGATCGGAAGAGC -B GTGACTGGAGTTCAGACGTGTGC -b AAGATCGGAAGAGCACACGTCTGAACTCCAGTCACNNNNNNNATCTCGTATGCCGTCTTCTGC TTG -A T{100} -A A{100} –adapter AAGATCGGAAGAGCACACGTCTGAACTCCAGTCAC NNNNNNNATCTCGTATGCCGTCTTCTGCTTG -A GATCGTCGGACTGTAGAACTCTGAACGTGTAGA –interleaved-input - –interleaved-output - –quiet –report-file /dev/stderr -f fastq.

The trimmed reads were first aligned against the UniVec database (https://www.ncbi.nlm.nih.gov/ tools/vecscreen/uvcurrent/#Replist) by BOWTIE2 (*Langmead and Salzberg, 2012*) to filter out common DNA contaminants, including sequencing adapters and commercial cloning vectors (Pass 1). The remaining reads were then aligned by BOWTIE2 to a customized reference containing rRNA (GenBank accession numbers: X12811.1 and U13369.1) and mitochondrial genome sequences (chrM from hg19) (Pass 2), followed by BOWTIE2 alignment to a small non-coding RNA reference containing sequences of tRNAs (GtRNAdb, http://gtrnadb.ucsc.edu), 7SK RNA, 7SL RNA, Y-RNAs, vault RNAs (Ensembl, https://useast.ensembl.org/index.html) and high confidence miRNAs (http://www. mirbase.org) (Pass 3). The remaining unaligned reads were then aligned against the human genome reference sequence (hg19) sequentially with HISAT2 (*Kim et al., 2019*) and BOWTIE2 (Pass 4). For transcriptome mapping, reads from Pass three that did not correspond to sncRNA, tRNA, rRNA, or

repeat sequences were mapped to human cDNA sequences (Ensemble hg19 cDNA references) using BOWTIE2.

Deduplication of mapped reads was done by UMI, CIGAR string, and genome coordinates (*Quinlan, 2014*). To accommodate base-calling and PCR errors and non-templated nucleotides that may have been added to the 3' ends of cDNAs during TGIRT-seq library preparation, one mismatch in the UMI was allowed during deduplication, and fragments with the same CIGAR string, genomic coordinates (chromosome start and end positions), and UMI or UMIs that differed by one nucleotide were collapsed into a single fragment. The counts for each read were readjusted to overcome potential UMI saturation for highly-expressed genes by implementing the algorithm described in *Fu et al., 2011*, using sequencing tools (https://github.com/wckdouglas/sequencing_tools).

## Whole-genome sequencing simulation

A paired-end 75-nt simulated NextSeq library was generated by ART (*Huang et al., 2012*) (https://www.niehs.nih.gov/research/resources/software/biostatistics/art/index.cfm) with a mean fragment length of 160 nt using parameters: -ss NS50 -sam -p -l 75 f 5 m 160 s 10.

## Analysis of protein-coding gene reads

Reads that mapped to protein-coding genes were extracted and quantified by Kallisto (*Bray et al., 2016*) using a transcript reference generated by gffread (*Pertea and Pertea, 2020*) from the hg19 human genome reference sequence and GENCODE transcript annotations release 28. Gene expression profiles for human tissues were downloaded from the Human Protein Atlas (*Uhlén et al., 2015*). Six platelet RNA-seq datasets from healthy individuals were obtained from NCBI Sequence Read Archive (SRR5907423-SRR5907428) (*Campbell et al., 2018*) and TPM values were quantified by Kallisto using the hg19 reference sequence. The percentage of bases that mapped to different regions of protein-coding genes was computed by CollectRnaSeqMetrics from Picard (Broad Institute) using a genome BAM file generated by Kallisto. Gene body coverage was computed using RSeQC (*Wang et al., 2012*).

## Metagenomic analysis

Unmapped reads from datasets for DNase-treated plasma RNA (n = 15) or NaOH-treated plasma DNA (n = 4) were combined with reads that mapped to the *E. coli* genome from read-mapping Pass 1 (see above) and analyzed by Kraken2 (*Wood et al., 2019*) using the miniKraken2 V2 database with default settings. The classification results were visualized by Pavian (*Breitwieser and Salzberg, 2020*). Mapping of the reads used for Kraken2 analysis to diatom reference sequences (*Akat et al., 2019*) showed low levels of diatom contamination (0.19%, 21,000 reads in DNase-treated plasma RNA (n = 15) and < 0.002% (22 reads) in NaOH-treated plasma DNA (n = 4)).

## Peak calling

For each sample, a BAM file with read pairs that mapped to the human-genome was processed as described above to remove vector sequences (UniVec, Pass 1), rRNAs and Mt RNAs (Pass 2), and annotated sncRNAs (a customized small non-coding RNA reference containing sequences of tRNAs [GtRNAdb, http://gtrnadb.ucsc.edu], 7SK RNA, 7SL RNA, Y-RNAs, vault RNAs [Ensembl, https://useast.ensembl.org/index.html], high confidence and non-high confidence miRNAs [http://www.mirbase.org], snoRNAs, snRNAs, and miscellaneous RNAs [Ensembl; https://useast.ensembl.org/index.html]). The alignment coordinates were extracted as a BED file and deduplicated as described above for sequence data processing. BED files from libraries with the same treatments were combined, and reads mapping to human genome blacklist regions (*Amemiya et al., 2019*) or Mt DNA were removed, the latter to eliminate mis-mapping of Mt RNAs to nuclear Mt DNA segments (NUMTs). The BED file for the combined DNase-treated datasets (n = 15) was then split into separate BED files containing only forward strand or reverse strand fragments. BED files from each strand were used as input the for MACS2 callpeak algorithm (*Zhang et al., 2008*) using the combined BED file for both strands from NaOH-treated plasma DNA datasets (n = 4) as the base-line control, via: macs2 callpeak –treatment {input_bed_strand_bed} –control {alkaline_hydrolysis_bed} –outdir {outdir} –name {sample_strand} –nomodel –format BEDPE –keep-dup all –gsize hs –broad. Peak calling against the human transcriptome (GENCODE release 28) was done by using the

following parameters: macs2 callpeak –treatment {input_bed_strand_bed} –outdir {outdir} –name {sample_strand} –nomodel –format BEDPE –keep-dup all –gsize hs –broad. The called peaks were filtered to obtain candidate peaks with read coverage (pileup) $\geq$ 5 at the peak maximum, a false discovery rate $\leq$ 0.05 (q-value assigned by MACS2), and a requirement that the peak be detected in at least 5 of the 15 samples to avoid batch effects. The candidate peaks were then further filtered to remove reads with MAPQ < 30 (to exclude reads that mapped equally well by BOWTIE2 to more than one locus) or had $\geq$ 5 mismatches from the mapped locus to obtain high confidence peaks with $\geq$ 5 high quality read alignments at the peak maximum. The remainder were classified as low confidence peaks (*Supplementary file 1*).

The annotation of peaks was done by BEDTools (*Quinlan, 2014*) intersect command against genomic features from RefSeq genes (https://www.ncbi.nlm.nih.gov/refseq/), Ensembl genes (https://useast.ensembl.org/index.html), Repbase repeat regions (*Bao et al., 2015*), RBP-binding site annotations from ENCODE eCLIP datasets (https://www.encodeproject.org/eclip/), and piRNA annotations from piRNAdb (https://www.pirnadb.org). An overlap score defined as the product of $\frac{Number\ of\ overlapping\ base}{Peak\ width}$ and $\frac{Number\ of\ overlapping\ bases}{Genomic\ feature\ size}$ was computed for each overlapping record between a peak and a genomic feature. An overlap score of 1 (highest possible overlap score) indicates the peak can be explained by the full-length genomic feature. A best feature annotation for each peak was selected by the highest overlap score. In cases of identical overlap scores, a feature was selected with the priority RBP-binding sites, repeat regions (Repbase), and lastly long RNAs. In cases of peaks overlapping with two or more RBP binding sites with same overlap score, all RBPs were reported. A histogram showing the number of overlapping bases between peaks and annotated eCLIP-identified RBP-binding sites is shown in *Figure 4—figure supplement 2B*.

## RNA homology search and RNA secondary structure prediction

RNA homology searches and secondary structure predictions were done using the hg19 genomic coordinates of the peak called by MACS2 (narrow peak). For long RNA, antisense, and unannotated peaks, we performed an RNA homology search against the covariance model from Rfam (*Kalvari et al., 2018*) using Infernal (*Nawrocki and Eddy, 2013*), via: cmscan -o {out_file} –tblout {out_table} –cpu 24 Rfam.cm peak.fa. Infernal/Rfam analysis for known structured RNA motifs was conducted by submitting the peak sequences to Rfam website (http://rfam.xfam.org). RNA secondary structures were predicted and MFEs calculated by RNAfold with default parameters using the ViennaRNA package (*Lorenz et al., 2011*).

## Empirical Bayes analysis of strand specificity of repeat regions

We noticed that peaks for some simple repeat sequences (*e.g.*, (CATTC)n and (TTAGGG)n) were called by MACS2 on both strands, but in different ratios in the datasets for DNase-treated plasma RNA and NaOH-treated plasma DNA. To identify repeat sequences with a significantly higher counts in one orientation over the other and eliminate ambiguous read mappings, we aggregated the strand-specific fragment counts for all unique simple repeat sequences and their reverse-complements (*e.g.*, (ATCCC)n and (GGGTA)n) in the combined DNase-treated plasma RNA (n = 15) and NaOH-treated plasma DNA (n = 4) datasets. To compensate for repeat sequences with low read counts, which could give misleading strand-specificity, we employed an Empirical Bayes method by using repeat sequences with high total counts (> 50 deduplicated reads) regardless of strand from the combined plasma DNA datasets to construct a prior beta distribution of strand specificities for all repeat sequences (*Figure 8C*). This fitted beta distribution represents the probability distribution of detecting 0–100% (+) orientation reads for each repeat sequence in a pure DNA sample. Using the fitted beta distribution hyperparameters and the count data from either the plasma DNA or plasma RNA datasets, we then computed a posterior beta-binomial distributions of the percentages of (+) orientation reads for each repeat sequences, which represents how likely and to what degree the strand specificities were biased towards the (+) or (-) strand orientation. We then compared the posterior distributions for each unique repeat sequences between plasma RNA and DNA to obtain a distribution of differences in the percentage of (+) strand reads in plasma RNA versus plasma DNA ($\Delta$ %(+) in RNA vs DNA). Summary statistics were then extracted from this final distribution, and repeat sequences with Bayes factor > 3 for at least 10% more (+) strand fragments in the plasma RNA compared to the plasma DNA samples were identified as significantly enriched in (+)-strand

fragments in plasma RNA. The Bayesian statistical testing framework was implemented using pymc3 (*Salvatier et al., 2016*).

## Code availability

All scripts used for data processing are deposited in GitHub: https://github.com/wckdouglas/cfNA. (*Wu and Yao, 2020*; copy archived at https://github.com/elifesciences-publications/cfNA).

## Date deposition

The TGIRT-seq datasets in this manuscript are listed in the Supplementary File and have been deposited in the National Center for Biotechnology Information Sequence Read Archive (accession number: PRJNA640428).

## Acknowledgements

This work was supported by NIH grants R01 GM37949 and R35 GM136216 and Welch Foundation grant F-1607. We thank Manny Ares (University of California, Santa Cruz), Sean Eddy (Harvard), and Eckhard Jankowsky (Case Western Reserve University) for helpful discussions. We also thank our colleagues Elizabeth Kiddie and Hengyi Xu for comments on the manuscript and Hengyi Xu for exonuclease I treatment of plasma nucleic acid samples. The Texas Advanced Computing Center (TACC) and the Center for Biomedical Research Support (CBRS) at the University of Texas at Austin provided high-performance computing resources. High-throughput sequencing was done by the Genomic Sequencing and Analysis Facility at the University of Texas at Austin.

## Additional information

### Competing interests

Jun Yao: is an inventor on a patent application filed by the University of Texas at Austin for the use of full-length excised intron RNAs and intron RNA fragments as biomarkers. US patent application 63/014,429. Douglas C Wu: is an inventor on a patent application filed by the University of Texas at Austin for the use of full-length excised intron RNAs and intron RNA fragments as biomarkers. US patent application 63/014,429; is currently an employee of QIAGEN. Alan M Lambowitz: Thermostable group II intron reverse transcriptase (TGIRT) enzymes and methods for their use are the subject of patents and patent applications that have been licensed by the University of Texas and East Tennessee State University to InGex, LLC. Is a minority equity holder in InGex, LLC and receives royalty payments from the sale of TGIRT-enzymes and kits and from the sublicensing of intellectual property by InGex to other companies. Is an inventor on a patent application filed by the University of Texas at Austin for the use of full-length excised intron RNAs and intron RNA fragments as biomarkers. US patent application 63/014,429. The other author declares that no competing interests exist.

### Funding

| Funder | Grant reference number | Author |
| --- | --- | --- |
| National Institute of General Medical Sciences | R01 GM37949 | Alan M Lambowitz |
| National Institute of General Medical Sciences | R35 GM136216 | Alan M Lambowitz |
| Welch Foundation | F-1607 | Alan M Lambowitz |

The funders had no role in study design, data collection and interpretation, or the decision to submit the work for publication.

### Author contributions

Jun Yao, Conceptualization, Data curation, Formal analysis, Validation, Investigation, Visualization, Methodology, Writing - original draft, Writing - review and editing; Douglas C Wu, Conceptualization, Data curation, Software, Formal analysis, Validation, Investigation, Visualization, Methodology,

Writing - original draft, Writing - review and editing; Ryan M Nottingham, Data curation, Investigation, Visualization, Writing - review and editing; Alan M Lambowitz, Conceptualization, Resources, Data curation, Funding acquisition, Visualization, Writing - original draft, Project administration, Writing - review and editing

Author ORCIDs
Jun Yao ⓘ https://orcid.org/0000-0002-1232-1587
Douglas C Wu ⓘ https://orcid.org/0000-0001-6179-3110
Ryan M Nottingham ⓘ http://orcid.org/0000-0001-6937-5394
Alan M Lambowitz ⓘ https://orcid.org/0000-0001-6036-2423

Decision letter and Author response
Decision letter https://doi.org/10.7554/eLife.60743.sa1
Author response https://doi.org/10.7554/eLife.60743.sa2

## Additional files

### Supplementary files
• Supplementary file 1. Comprehensive peak annotations and supporting data.

• Transparent reporting form

### Data availability
Code availability: All scripts used for data processing are deposited in GitHub: https://github.com/wckdouglas/cfNA (copy archived at https://github.com/elifesciences-publications/cfNA) Date deposition: The TGIRT-seq datasets in this manuscript are listed in the Supplementary File and have been deposited in the National Center for Biotechnology Information Sequence Read Archive (accession number: PRJNA640428).

The following dataset was generated:

| Author(s) | Year | Dataset title | Dataset URL | Database and Identifier |
|---|---|---|---|---|
| Yao J, Wu D, Nottingham R, Lambowitz AM | 2020 | Identification of protein-protected mRNA fragments and structured excised intron RNAs in human plasma by TGIRT-seq peak calling | https://www.ncbi.nlm.nih.gov/sra/PRJNA640428 | NCBI Sequence Read Archive, PRJNA640428 |

The following previously published datasets were used:

| Author(s) | Year | Dataset title | Dataset URL | Database and Identifier |
|---|---|---|---|---|
| Campbell RA, Franks Z, Bhatnagar A, Rowley JW, Manne BK, Supiano MA, Schwertz H, Weyrich AS, Rondina MT | 2017 | Granzyme A in Human Platelets Regulates Pro-Inflammatory Gene Synthesis by Monocytes in Aging | https://www.ncbi.nlm.nih.gov/bioproject/PRJNA397446 | NCBI BioProject, PRJNA397446 |
| Qin Y, Yao J, Wu DC, Nottingham RM, Mohr S, Hunicke-Smith S, Lambowitz AM | 2015 | TGIRT-Seq profiling of human plasma nucleic acids | https://www.ncbi.nlm.nih.gov/bioproject/PRJNA297566 | NCBI BioProject, PRJNA297566 |

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
