## [Decision Letter]

**Acceptance summary:**

In this paper the authors use a thermostable reverse transcriptase encoded by a so-called group II intron to sequence RNAs present in blood. This reverse transcriptase differs from retroviral reverse transcriptases in that it can read through structured regions of RNA. Results presented indicate that the population of RNAs in blood is more complex than previously thought.

**Decision letter after peer review:**

Thank you for submitting your article "Identification of protein-protected mRNA fragments and structured intron RNAs in human plasma by TGIRT-seq peak calling" for consideration by *eLife*. Your article has been reviewed by three peer reviewers, and the evaluation has been overseen by Timothy Nilsen as Reviewing Editor and James Manley as the Senior Editor. The following individuals involved in review of your submission have agreed to reveal their identity: Mariano A Garcia-Blanco (Reviewer #1).

The reviewers have discussed the reviews with one another and the Reviewing Editor has drafted this decision to help you prepare a revised submission.

The reviewers agreed that the work was, in principle, suitable for *eLife*. Nevertheless, each raised significant concerns that need to be addressed via revision. Specifically, the reviewers felt that the bioinformatic analysis needs to be strengthened. They also requested that you provide some practical application of the work. Please address all of the points raised by the reviewers as thoroughly as possible.

Reviewer #1:

The Lambowitz group has developed thermostable group II intron reverse transcriptases (TGIRTs) that strand switch and also have trans-lesion activity to provide a much wider view of RNA species analyzed by massively parallel RNA sequencing. In this manuscript they use several improvements to their methodology to identify RNA biotypes in human plasma pooled from several healthy individuals. Additionally, they implicate binding by proteins (RBPs) and nuclease-resistant structures to explain a fraction of the RNAs observed in plasma. Generally, I find the study fascinating and argue that the collection of plasma RNAs described is an important tool for those interested in extracellular RNAs. I think the possibility that RNPs are protecting RNA fragments in circulation is exciting and fits with elegant studies of insects and plants where RNAs are protected by this mechanism and are transmitted between species.

I have one major comment for the authors to consider. In my view the use of pooled plasma samples prevented the important opportunity to provide a glimpse on human variation in plasma RNA biotypes. This significantly limits the use of this information to begin addressing RNA biotypes as biomarkers. While I realize that data from multiple individuals represents a significant undertaking and may be beyond the scope of this manuscript, I urge the authors to do two things: (1) downplay the significance of the current study on the development of biomarkers in the current manuscript (e.g., in the Abstract and Discussion – e.g., "The ability of TGIRT-seq to simultaneously profile a wide variety of RNA biotypes in human plasma, including structured RNAs that are intractable to retroviral RTs, may be advantageous for identifying optimal combinations of coding and non-coding RNA biomarkers for human diseases."). (2) Carry out an analysis in multiple individuals – including racially diverse individuals – very important information will come of this – similar to C. Burge's important study in Nature 2008 where it was clear that there is important individual variation in alternative splicing decisions – very likely genetically determined. This second suggestion could be added here or constitute a future manuscript.

Reviewer #2:

Yao et al. used thermostable group II intron reverse transcriptase sequencing (TGIRT-seq) to study apheresis plasma samples. The first interesting discovery is that they had identified a number of mRNA reads with putative binding sites of RNA-binding proteins. A second interesting discovery from this work is the detection of full-length excised intron RNAs.

I have the following comments.

1) One doubt that I have is how representative is apheresis plasma when compared with plasma that one obtains through routine centrifugation of blood. The authors have reported the comparison of apheresis plasma versus a single male plasma in a previous publication. I think that to address this important question, a much-increased number of samples would be necessary.

2) For the important conclusion of the presence of binding sites of RNA-binding proteins in a proportion of apheresis plasma mRNA molecules, the authors need to explore whether there is any systemic difference in terms of mapping quality (i.e. mapping quality scores in alignment results) between RBP binding sites and non-RBP binding sites, so that any artifacts of peaks caused by the alignment issues occurring in RNA-seq analysis could be revealed and solved subsequently. Furthermore, it would be prudent to perform immunoprecipitation experiments to confirm this conclusion in at least a proportion of the mRNA.

3) In Figure 2D, one can observe that there are clearly more RNA reads in TGIRT-seq located in the 1st exon of ACTB, compared with SMART-seq. Is there any explanation? Will this signal be called as a peak (a potential RBP binding site) in the peak calling analysis (MACS2)? Is ACTB supposed to be bound by a certain RBP?

4) For Figure 2A, it would be informative for the comparison of RNA yield and RNA size profile among different protocols if the author also added the results of TGIRT-seq.

5) As shown in Figure 4C (the track of RBP binding sites), it seems quite pervasive in some gene regions. How many RBP binding sites from public eCLIP-seq results are used for overlapping peaks present in TGIRT-seq of plasma RNA? What percentage of plasma RNA reads are fallen within RBP binding sites? Are those peaks present in TGRIT-seq significantly enriched in RBPs binding regions?

6) Since there is a considerable portion of TGIRT-seq reads related to simple repeat, one possible reason is likely the high abundance of endogenous repeat-related RNA species in plasma. Nonetheless, have authors studied whether the ligation steps in TGIRT-seq have any biases (e.g. GC content) when analyzing human reference RNAs and spike ins (Introduction paragraph three)?

7) As described in Figure 2 legend, there are 0.25 million deduplicated reads for TGIRT-seq reads assigned to protein-coding genes transcripts which are far less than 2.18 million reads for SMART-seq. The authors need to discuss whether the current protocol of TGIRT-seq would cause potential dropouts in mRNA analysis, compared with SMART-seq?

8) While scientific thought-provoking, the practical implication of the current work is still unclear. The authors have suggested that their work might have applications for biomarker development. Is it possible to provide one experimental example in the manuscript?

Reviewer #3:

In this work, Yao and colleagues described transcriptome profiling of human plasma from healthy individuals by TGIRT-seq. TGIRT is a thermostable group II intron reverse transcriptase that offers improved fidelity, processivity and strand-displacement activity, as compared to standard retroviral RT, so that it can read through highly structured regions. Similar analysis was performed previously (Qin et al., 2016), but this study incorporated several improvements in library preparation including optimization of template switching condition and modified adapters to reduce primer dimer and introduce UMI. In their analysis, the authors detected a variety of structural RNA biotypes, as well as reads from protein-coding mRNAs, although the latter is in low abundance. Compared to SMART-Seq, TGIRT-seq also achieved more uniform read coverage across gene bodies. One novel aspect of this study is the peak analysis of TGIRT-seq reads, which revealed ~900 peaks over background. The authors found that these peaks frequently overlap with RBP binding sites, while others tend to have stable predicted secondary structures, which explains why these regions are protected from degradation in plasma. Overall, this study provided a robust dataset and expanded picture of RNA biotypes one can detect in human plasma. This is valuable because the findings may have implications in biomarker identification in disease contexts. On the other hand, the manuscript, in the current form, is relatively descriptive, and can be improved with a clearer message of specific knowledge that can be extracted from the data.

Specific points:

1) Several aspects of bioinformatics analysis can be clarified in more detail. For example, it is unclear how sequencing errors in UMI affect their de-duplication procedure. This is important for their peak analysis, so it should be explained clearly. Also, it is not described how exon junction reads (when mapped to the genome) are handled in peak calling, although the authors did perform complementary analysis by mapping reads to the reference transcriptome.

2) Overall, the authors provided convincing data that TGIRT-seq has advantages in detecting a wide range of RNA biotypes, especially structured RNAs, compared to other protocols, but these data are more confirmatory, rather than completely new findings (e.g., compared to Qin et al., 2016).

3) The peak analysis is more novel. The authors observed that 50% of peaks in long RNAs overlap with eCLIP peaks. However, there is no statistical analysis to show whether this overlap is significant or simply due to the pervasive distribution of eCLIP peaks. In fact, it was reported by the original authors that eCLIP peaks cover 20% of the transcriptome. Similarly, the authors found that a high proportion of remaining peaks can fold into stable secondary structures, but this claim is not backed up by statistics either.

4) Ranking of RBPs depends on the total number of RBP binding sites detected by eCLIP, which is determined by CLIP library complexity and sequencing depth. This issue should be at least discussed.

5) Enrichment of RBP binding sites and structured RNA in TGIRT-seq data is certainly consistent with one's expectation. However, the paper can be greatly improved if the authors can make a clearer case what is new that can be learned, as compared to eCLIP data or other related techniques that purify and sequence RNA fragments crosslinked to proteins. What is the additional, independent evidence to show the predicted secondary structures are real?

6) The authors should probably discuss how alignment errors can potentially affect detection of repetitive regions.

7) Many figures are IGV screenshots, which can be difficult to follow. Some of them can probably be summarized to deliver the message better.

---

## [Author Response]

Reviewer #1:[…]I have one major comment for the authors to consider. In my view the use of pooled plasma samples prevented the important opportunity to provide a glimpse on human variation in plasma RNA biotypes. This significantly limits the use of this information to begin addressing RNA biotypes as biomarkers. While I realize that data from multiple individuals represents a significant undertaking and may be beyond the scope of this manuscript, I urge the authors to do two things: (1) downplay the significance of the current study on the development of biomarkers in the current manuscript (e.g., in the Abstract and Discussion – e.g., "The ability of TGIRT-seq to simultaneously profile a wide variety of RNA biotypes in human plasma, including structured RNAs that are intractable to retroviral RTs, may be advantageous for identifying optimal combinations of coding and non-coding RNA biomarkers for human diseases."). (2) Carry out an analysis in multiple individuals – including racially diverse individuals – very important information will come of this – similar to C. Burge's important study in Nature 2008 where it was clear that there is important individual variation in alternative splicing decisions – very likely genetically determined. This second suggestion could be added here or constitute a future manuscript.

The identification of biomarkers in human plasma is an important application of this study, as was noted by reviewer 3: "Overall, this study provided a robust dataset and expanded picture of RNA biotypes one can detect in human plasma. This is valuable because the findings may have implications in biomarker identification in disease contexts." The present manuscript lays the foundation for such applications, which we have been carrying out in parallel. In one such study in collaboration with Dr. Naoto Ueno (MD Anderson), we used TGIRT-seq to identify combinations of mRNA and non-coding RNA biomarkers in FFPE-tumor slices, PBMCs and plasma from inflammatory breast cancer patients compared to non-IBC breast cancer patients and healthy controls (manuscript in preparation; data presented publicly in seminars), and in another, we explored the potential of using full-length excised intron (FLEXI) RNAs as biomarkers. In the latter study, we identified > 8,000 FLEXI RNAs in different human cell lines and tissues and found that they are expressed in a cell-type specific manner, including hundreds of differences between matched tumor and healthy tissues from breast cancer patients and cell lines. A manuscript describing these findings has been posted on bioRxiv (Yao et al., 2020, https://doi.org/10.1101/2020.09.07.285114). This new manuscript follows directly from the last sentence of the present manuscript and fully references the bioRxiv preprint currently under review for *eLife*.

Reviewer #2:[…]1) One doubt that I have is how representative is apheresis plasma when compared with plasma that one obtains through routine centrifugation of blood. The authors have reported the comparison of apheresis plasma versus a single male plasma in a previous publication. I think that to address this important question, a much-increased number of samples would be necessary.

Detailed comparison of plasma prepared by apheresis to that prepared by centrifugation would require a separate large-scale study, preferably by multiple laboratories using different methods to prepare plasma. However, our impression both from our findings and from the literature (Valbonesi et al., 2001, cited in the manuscript) is that apheresis-prepared plasma has very low levels of cellular contamination (required to meet clinical standards) compared to plasma prepared by centrifugation, even with protocols designed to minimize contamination from intact or broken cell (e.g., preparing plasma from freshly drawn blood, centrifugation into a Ficoll cushion to minimize cell breakage, and carefully avoiding contamination from sedimented cells).

We do have additional information about the degree of variation in protein-coding gene transcripts detected by TGIRT-seq in plasma samples prepared by centrifugation from five healthy females controls in our collaborative study with Dr. Naoto Ueno (M.D. Anderson; see above), and we have added it to the manuscript citing a manuscript in preparation with permission from Dr. Ueno as follows:

“The identities and relative abundances of different protein-coding gene transcripts in the apheresis-prepared plasma were broadly similar to those in the previous TGIRT analysis of plasma prepared by Ficoll-cushion sedimentation of blood from a healthy male individual (Qin et al., 2016) (r = 0.62-0.80; Figure 3C) and between high quality plasma samples similarly prepared from five healthy females in a collaborative study with Dr. Naoto Ueno, M.D. Anderson (r = 0.53-0.67; manuscript in preparation).” See Author response image 1.

**Author response image 1. sa2fig1:** Correlation matrix of protein coding genes detected in DNase I-treated pooled plasma RNA using the improved TGIRT-seq method in this work (n=12), previously by TGIRT-seq in DNase I-treated plasma RNA from a healthy male individual (n=3; SRP064378, datasets 12-14) (Qin et al., 2016), and healthy plasmas samples similarly prepared from five healthy females in a collaborative study with Dr. Naoto Ueno, M.D. Anderson. For comparisons, CPMs for each gene in a dataset were normalized to the total number of mapped reads in that dataset, log_2_ transformed, and plotted. Spearman correlation coefficients are shown on the upper triangle. Numbers on the x and y axes of scatter plots are log_2_ transformed CPMs.

2) For the important conclusion of the presence of binding sites of RNA-binding proteins in a proportion of apheresis plasma mRNA molecules, the authors need to explore whether there is any systemic difference in terms of mapping quality (i.e. mapping quality scores in alignment results) between RBP binding sites and non-RBP binding sites, so that any artifacts of peaks caused by the alignment issues occurring in RNA-seq analysis could be revealed and solved subsequently. Furthermore, it would be prudent to perform immunoprecipitation experiments to confirm this conclusion in at least a proportion of the mRNA.

We have added a figure panel comparing MAPQ scores for reads from peaks containing RBP-binding sites to those of other long RNA reads (Figure 4—figure supplement 2A) and have added further details about the methods used to obtain peaks with high quality reads, including the following:

“After further filtering to remove read alignments with MAPQ < 30 (a cutoff that eliminates reads mapping equally well at more than one locus) or ≥ 5 mismatches from the mapped locus, we were left with 950 high confidence peaks ranging in size from 59 to 1,207 nt with ≥ 5 high quality read alignments at the peak maximum (Supplementary file 1).”

3) In Figure 2D, one can observe that there are clearly more RNA reads in TGIRT-seq located in the 1st exon of ACTB, compared with SMART-seq. Is there any explanation? Will this signal be called as a peak (a potential RBP binding site) in the peak calling analysis (MACS2)? Is ACTB supposed to be bound by a certain RBP?

The higher coverage of the ACTB 5'-exon in the TGIRT-seq datasets reflects in part the more uniform 5' to 3' coverage of mRNA sequences by TGIRT-seq compared to SMART-seq, which is biased for 3'-mRNA sequences that have poly(A) tails (current Figure 3F). The signal in the first exon of ACTB was in fact called as a peak by MACS2 (peak ID#893, Supplementary file 1), which overlapped an annotated binding site for SERBP1 (see Supplementary file 1).

4) For Figure 2A, it would be informative for the comparison of RNA yield and RNA size profile among different protocols if the author also added the results of TGIRT-seq.

Figure 3D (previously Figure 2A) shows a bioanalyzer trace of PCR amplified cDNAs obtained by SMART-Seq. These cDNAs correspond to 3' mRNA sequences that have poly(A) tails and are not comparable to the bioanalyzer profiles of plasma RNA (Figure 1—figure supplement 1) or read span distributions in the TGIRT-seq datasets (Figure 1B), which are dominated by sncRNAs. The normalized 5' to 3' gene body coverage plots for protein-coding gene transcripts show that TGIRT-seq captures mRNA fragments that collectively span the entire mRNA sequence, whereas SMART-Seq is biased for 3' sequences linked to poly(A) (Figure 3F). We also note that the normalized gene body coverage plots and mRNAs detected by TGIRT-seq remain similar, even if the plasma RNA is chemically fragmented prior to TGIRT-seq library construction (Figure 3F and Figure 3—figure supplement 2).

5) As shown in Figure 4C (the track of RBP binding sites), it seems quite pervasive in some gene regions. How many RBP binding sites from public eCLIP-seq results are used for overlapping peaks present in TGIRT-seq of plasma RNA? What percentage of plasma RNA reads are fallen within RBP binding sites? Are those peaks present in TGRIT-seq significantly enriched in RBPs binding regions?

We noted that 109 RBP-binding sites were searched in the initial analysis, and we have now added further analyses for 150 RBPs currently available in ENCODE eCLIP datasets with irreproducible discovery rate (IDR) analysis (Figure 6 and Figure 6—figure supplement 1). We have also added a tab to the Supplementary file 1 identifying the 109 and 150 RBPs whose binding sites were searched. The requested statistical analysis has been added in Figure 4—figure supplement 2C. The analysis shows that enrichment of RBP-binding site sequences in the 467 identified RBP-binding site peaks was statistically significant (p < 0.001) (subsection “RNA fragments containing RBP-binding sites are prevalent in plasma”).

6) Since there is a considerable portion of TGIRT-seq reads related to simple repeat, one possible reason is likely the high abundance of endogenous repeat-related RNA species in plasma. Nonetheless, have authors studied whether the ligation steps in TGIRT-seq have any biases (e.g. GC content) when analyzing human reference RNAs and spike ins (Introduction paragraph three)?

We have added a note to the manuscript indicating that although repeat RNAs constitute a high proportion of the called peaks, they do not constitute a similarly high proportion of the total RNA reads (Figure 1C; subsection “Identification of tandem repeats and transposable element RNAs”, first sentence). The TGIRT-seq analysis of human reference RNAs and spike-ins showed that TGIRT-seq recapitulates the relative abundance of human transcripts and spike-in comparably to non-strand-specific TruSeq-v2 and better than strand-specific TruSeq-v3 (Nottingham et al., 2016). Additionally, we used miRNA reference sets for detailed analysis of TGIRT-seq biases, including developing a computer algorithm for bias correction based on a random forest regression model that provides insight into different factors that contribute to these biases (Xu et al., 2019). Overall GC content does not make a significant contribution to TGIRT-seq biases (Figure 9 of Xu et al., 2019). Instead, biases in TGIRT-seq are largely confined to the first three nucleotides at the 5'-end (due to bias of the thermostable 5' App DNA ligase used for 5' RNA-seq adapter addition) and the 3' nucleotide (due to TGIRT-template switching). These end biases are not expected to significantly impact the quantitation of repeat RNAs.

7) As described in Figure 2 legend, there are 0.25 million deduplicated reads for TGIRT-seq reads assigned to protein-coding genes transcripts which are far less than 2.18 million reads for SMART-seq. The authors need to discuss whether the current protocol of TGIRT-seq would cause potential dropouts in mRNA analysis, compared with SMART-seq?

We have added the following to the manuscript:

“The larger number of mRNA reads compared to TGIRT-seq (0.28 million) largely reflects that SMART-seq selectively profiles polyadenylated mRNAs, while TGIRT-seq profiles mRNAs together with other more abundant RNA biotypes. In addition, ultra-low input SMART-Seq is not strand-specific, resulting in redundant sense and antisense strand reads (Figure 3—figure supplement 1).”

The manuscript contains the following statement regarding potential dropouts:

“A scatter plot comparing the relative abundance of transcripts originating from different genes showed that most of the polyadenylated mRNAs detected in DNase I-treated plasma RNA by ultra-low input SMART-Seq were also detected by TGIRT-seq at similar TPM values when normalized for protein-coding gene reads (*r* = 0.61), but with some, mostly lower abundance mRNAs undetected either by TGIRT-seq or SMART-Seq, and with SMART-Seq unable to detect non-polyadenylated histone mRNAs, which are relatively abundant in plasma (Figure 3E and Figure 3—figure supplement 1).”

8) While scientific thought-provoking, the practical implication of the current work is still unclear. The authors have suggested that their work might have applications for biomarker development. Is it possible to provide one experimental example in the manuscript?

We addressed the relevance of the manuscript to biomarker identification and noted parallel studies that supports this application in the response to reviewer 1, comment 1. We have also modified the final paragraph of the Discussion:

“The ability of TGIRT-seq to simultaneously profile a wide variety of RNA biotypes in human plasma, including structured RNAs that are intractable to retroviral RTs, may be advantageous for identifying optimal combinations of coding and non-coding RNA biomarkers that could then be incorporated in target RNA panels for diagnosis and routine monitoring of disease progression and response to treatment. […] Finally, we note that in addition to their biological and evolutionary interest, short full-length excised intron RNAs and intron RNA fragments, such as those identified here, may be uniquely well suited to serve as stable RNA biomarkers, whose expression is linked to that of numerous protein-coding genes."

Reviewer #3:[…]1) Several aspects of bioinformatics analysis can be clarified in more detail. For example, it is unclear how sequencing errors in UMI affect their de-duplication procedure. This is important for their peak analysis, so it should be explained clearly.

We have added details of the procedure used for de-duplication to the following paragraph in Materials and methods:

“Deduplication of mapped reads was done by UMI, CIGAR string, and genome coordinates (Quinlan, 2014). […] The counts for each read were readjusted to overcome potential UMI saturation for highly-expressed genes by implementing the algorithm described in (Fu et al., 2011), using sequencing tools (https://github.com/wckdouglas/sequencing_tools).”

Also, it is not described how exon junction reads (when mapped to the genome) are handled in peak calling, although the authors did perform complementary analysis by mapping reads to the reference transcriptome.

We have added this to first sentence of the paragraph describing peak calling against the transcriptome reference, which now reads as follows:

"Peak calling against the human genome reference sequence might miss RBP-binding sites that are close to or overlap exon junctions, as such reads were treated by MACS2 as long reads that span the intervening intron."

2) Overall, the authors provided convincing data that TGIRT-seq has advantages in detecting a wide range of RNA biotypes, especially structured RNAs, compared to other protocols, but these data are more confirmatory, rather than completely new findings (e.g., compared to Qin et al., 2016).

We modified the first paragraph of the Discussion to explicitly describe what is added by the present manuscript compared to Qin et al., 2016. Additionally, further analysis in response to the reviewers' comments resulted in the interesting finding that stress granule proteins comprised a high proportion of the RBPs whose binding sites were enriched in plasma RNAs (to our knowledge a completely new finding), consistent with a previously suggested link between RNP granules, EV packaging, and RNA export (paragraph five subsection “RNA fragments containing RBP-binding sites are prevalent in plasma”; data shown in Figure 6 and Figure 6—figure supplement 1). Also highlighted in the Discussion paragraph five.

3) The peak analysis is more novel. The authors observed that 50% of peaks in long RNAs overlap with eCLIP peaks. However, there is no statistical analysis to show whether this overlap is significant or simply due to the pervasive distribution of eCLIP peaks. In fact, it was reported by the original authors that eCLIP peaks cover 20% of the transcriptome.

We have added statistical analysis, which shows that the enrichment of RBP-binding sites in the 467 called peaks is statistically significant at p < 0.001 (paragraph one subsection “RNA fragments containing RBP-binding sites are prevalent in plasma”; Figure 4—figure supplement 2C), as well as scatter plots identifying proteins whose binding sites were more highly represented in plasma than cellular RNAs or vice versa (paragraph five subsection “RNA fragments containing RBP-binding sites are prevalent in plasma”; Figure 6 and Figure 6—figure supplement 1).

Similarly, the authors found that a high proportion of remaining peaks can fold into stable secondary structures, but this claim is not backed up by statistics either.

First, near the beginning of the paragraph describing these findings, we added the following to provide a guide as to what can and can't be concluded by RNAfold:

"To evaluate whether these peaks contained RNAs that could potentially fold into stable secondary structures, we used RNAfold, a tool that is widely used for this purpose with the understanding that the predicted structures remain to be validated and could differ under physiological conditions or due to interactions with proteins."

Second, at the end of the same paragraph, we have added the requested statistics:

"Subject to the caveats above regarding conclusions drawn from RNAfold, simulations using peaks randomly generated from long RNA gene sequences indicated that enrichment of RNAs with more stable secondary structures (lower MFEs) in the called RNA peaks was statistically significant (p≤0.019; Figure 4—figure supplement 2D)."

4) Ranking of RBPs depends on the total number of RBP binding sites detected by eCLIP, which is determined by CLIP library complexity and sequencing depth. This issue should be at least discussed.

We have added scatter plots in Figure 6 and Figure 6—figure supplement 1, which show that the relative abundance of different RBP-binding sites detected in plasma differs markedly from that for cellular RNAs in the eCLIP datasets (both for the 109 RBPs searched initially and for 150 RBPs withirreproducible discovery rate (IDR) analysis from the ENCODE web site). As mentioned in comments above, this analysis identified a number of RBP-binding sites that were substantially enriched in plasma RNAs compared to cellular RNAs or vice versa and led to what we think is the important new finding that plasma RNAs are enriched binding sites for a number of stress granule proteins (Figure 6 and Figure 6—figures supplement 1). We thank the reviewers for this and related comments that led to this additional analysis.

5) Enrichment of RBP binding sites and structured RNA in TGIRT-seq data is certainly consistent with one's expectation. However, the paper can be greatly improved if the authors can make a clearer case what is new that can be learned, as compared to eCLIP data or other related techniques that purify and sequence RNA fragments crosslinked to proteins. What is the additional, independent evidence to show the predicted secondary structures are real?

Compared to CLIP and related methods, peak calling enables more facile identification of candidate RBPs and putatively structured RNAs for further analysis and may be particularly useful for the vanishingly small amounts of RNA present in plasma and other bodily fluids. New findings resulting from peak calling in the present manuscript include that plasma RNAs are enriched in binding sites for stress granule proteins (see above) and the discovery of a variety of novel RNAs, including the full-length excised intron RNAs first identified here and subsequently studied in cellular RNAs in Yao et al. (https://doi.org/10.1101/2020.09.07.285114). We also note that peak calling enables the identification of protein-protected and structured mRNA regions that are relatively stable in plasma and may be more reliably detected in targeted liquid biopsy assays than are more labile mRNA regions.

6) The authors should probably discuss how alignment errors can potentially affect detection of repetitive regions.

In the Empirical Bayes method that we used for the analysis of repeats, repeat sequences were quantified by aggregate counts irrespective of the genomic locus to which they mapped (Materials and methods, subsection “Empirical Bayes analysis of strand specificity of repeat regions”), which should not be affected by alignment errors.

7) Many figures are IGV screenshots, which can be difficult to follow. Some of them can probably be summarized to deliver the message better.

Some IGV-based figures are crucial for showing key features of the RNAs that are called as peaks (e.g., the predicted secondary structures of the full-length excised intron RNAs and intron RNA fragments). However, in the process of reformatting, we have switched in and added non-IGV main text figures including Figure 2 (microbiome analysis), Figure 3 (TGIRT-seq versus SMART-Seq), Figure 4 (repeats), and Figure 6 (new figure comparing relative abundance of RBP-binding sites in plasma versus cells).